# Stress induced TDP-43 mobility loss independent of stress granules

Lisa Streit[1,2], Timo Kuhn[1], Thomas Vomhof [1], Verena Bopp[2], Albert C. Ludolph[2,3], Jochen H. Weishaupt[2,4], J. Christof M. Gebhardt [1], Jens Michaelis [1] ✉ & Karin M. Danzer [2,3] ✉

TAR DNA binding protein 43 (TDP-43) is closely related to the pathogenesis of amyotrophic lateral sclerosis (ALS) and translocates to stress granules (SGs). The role of SGs as aggregation-promoting "crucibles" for TDP-43, however, is still under debate. We analyzed TDP-43 mobility and localization under different stress and recovery conditions using live cell single-molecule tracking and super-resolution microscopy. Besides reduced mobility within SGs, a stress induced decrease of TDP-43 mobility in the cytoplasm and the nucleus was observed. Stress removal led to a recovery of TDP-43 mobility, which strongly depended on the stress duration. 'Stimulated-emission depletion microscopy' (STED) and 'tracking and localization microscopy' (TALM) revealed not only TDP-43 substructures within stress granules but also numerous patches of slow TDP-43 species throughout the cytoplasm. This work provides insights into the aggregation of TDP-43 in living cells and provide evidence suggesting that TDP-43 oligomerization and aggregation takes place in the cytoplasm separate from SGs.

TAR DNA binding protein 43 (TDP-43) is a predominantly nuclear DNA-/RNA-binding protein. Cytoplasmic TDP-43 inclusions are a neuropathological hallmark of amyotrophic lateral sclerosis (ALS) and frontotemporal lobar degeneration (FTD)[1,2]. ALS is a fatal neurodegenerative disease involving the progressive degeneration of the upper and lower motor neurons, leading to paralysis and death within 3–5 years after symptom onset[3]. FTD is an entity that is clinically and genetically linked to ALS and most often associated with behavioral alterations, personality changes and dementia. Besides the formation of pathological TDP-43 in many FTD and >95% of ALS patients[1,2], also the existence of ALS-causing TDP-43 mutations attributes and important role of TDP-43 for the pathogenesis of ALS/FTD[4].

Understanding the aggregation process is therefore broadly investigated and highly relevant for understanding the molecular pathogenesis of ALS/FTD. One starting point for the aggregation of TDP-43 could be its localization to and interaction with stress granules (SGs). SGs constitute a small reaction volume, densely packed with aggregation prone proteins and have been suspected to serve as a starting point for aggregation of several proteins related to neurodegeneration[5–7]. However, it is still under debate, whether TDP-43 localization to stress granules prevents or stimulates its pathological aggregation. Recent studies hint towards a mechanism where localization to stress granules and RNA interactions may exert, an initially protective rather than a disease-promoting role[8–11]. How TDP-43 mobility changes on a molecular level and in different cellular compartments and conditions is not known so far. It will be of central importance to understand if TDP-43 exhibits different mobility regimes depending on the spatial subcellular distribution.

In this work we therefore employed single-molecule live cell tracking, super-resolution stimulated emission depletion (STED) and

[1]Institute of Biophysics, Ulm University, Albert-Einstein-Allee 11, 89081 Ulm, Germany. [2]Neurology, University Clinic, Albert-Einstein-Allee 11, 89081 Ulm, Germany. [3]DZNE, Albert-Einstein-Allee 11, 89081 Ulm, Germany. [4]Division for Neurodegenerative Diseases, Neurology Department, Mannheim Center for Translational Medicine, University Medicine Mannheim, Heidelberg University, Mannheim, Germany. ✉e-mail: jens.michaelis@uni-ulm.de; karin.danzer@dzne.de

tracking and localization microscopy (TALM), to gain more insight into TDP-43 behavior and spatial distribution on a molecular level in living cells.

## Results

### Establishment of a model system for live cell single-molecule tracking of TDP-43

To study TDP-43 aggregation, its mobility in different cellular compartments and its role in stress granules on a single-molecule level, we engineered dual-transgenic TDP-43-Halo and G3BP1-SNAP cell lines (see Methods). As the attachment of bright, membrane permeable and photo-stable fluorophores is a key requisite for single-molecule studies, the commonly used HaloTag[12] and SNAP-Tag[13] systems were used in this study. To ensure that the fusion of the HaloTag (33 kDa) to TDP-43 does not alter the protein's localization and function, the HaloTag was fused to either the N- or the C-terminus of TDP-43, named $^{Halo}$TDP-43 and TDP-43$^{Halo}$, respectively (Fig. 1a, Supplementary Fig. 1a). To spatially assign stress granules during image analysis we used G3BP1$^{SNAP}$ as a stress granule marker.

We first ensured proper functionality of the tagged TDP-43 and G3BP1 constructs in the transgenic cell lines by Sir-Halo and TMR-SNAP staining, as well as immunostaining of endogenous proteins, and compared observed localizations as a function of stress duration to that in naïve H4 cells (Fig. 1b, c). In contrast to a diffusive cytoplasmic G3BP1 signal under unstressed conditions, G3BP1 positive stress granules containing TDP43$^{Halo}$ could be observed after 30 min and 60 min of sodium arsenite stress (Fig. 1d), which was also confirmed by immunostaining (Fig. 1e).

We then quantified the total of TDP-43$^{Halo}$ and G3BP1$^{SNAP}$ overexpressed proteins by Western blot analysis (Fig. 1f). Densitometric analysis showed an over-expression of 1.5–2 fold for the TDP-43$^{Halo}$ construct compared to endogenous TDP-43 in naïve H4 cells (Fig. 1g). We focused on C-terminally tagged TDP-43$^{Halo}$, since it allows for the visualization of both, full-length and fragmented TDP-43 species (Fig. 1f). Complementary data for the N-terminally tagged TDP-43 construct ($^{Halo}$TDP-43) are given in the supplementary materials (Supplementary Figs. 1 and 2).

C-terminally tagged TDP-43 showed the formation of a prominent 35 kDa TDP-43 fragment and an increased cytoplasmic localization, which was not seen for the N-terminally tagged TDP-43. Fragmentation of TDP-43 leads to the disruption or complete abolishment of the nuclear localization sequence (NLS)[14,15]. A disrupted or missing NLS leads to a subsequent accumulation of TDP-43 fragments in the cytoplasm, and could explain the observed slightly higher cytoplasmic level of TDP-43$^{Halo}$ as compared to the naïve H4 cells (Fig. 1c).

Taken together, the attachment of either Tag the respective protein, did not alter the formation of stress-induced phase-separated compartments or their interaction of G3BP1 and TDP-43 with latter. This establishes the transgenic cell lines as a model system for studying stress-induced dynamical changes of TDP-43 mobility using single-molecule imaging of Halo-tagged TDP-43 within different cellular compartments.

### Single-molecule tracking monitors the region-specific TDP-43 mobility

To monitor TDP-43 in a region-specific manner, we established a single molecule tracking pipeline, using photoactivatable Janelia Fluor 646 for labeling of Halo-tagged TDP-43[16] and TMR-labeling for G3BP1$^{SNAP}$ (Fig. 2a). The usage of a photoactivatable dye enabled labeling at high concentrations (nM range) and single-molecule tracking concentrations were achieved by continuously activating only a small subset of labeled TDP-43$^{Halo}$ per frame. Thus, a high number of frames could be recorded while continuously controlling the frame-wise emitter density[17,18].

TDP-43$^{Halo-PA-JF646}$ molecules were imaged for 120 min under unstressed or stressed conditions (0.5 mM sodium arsenite) using continuous 405 nm activation. In addition, every 200 frames (i.e. every 1.34 s) the G3BP1$^{SNAP-TMR}$ channel was recorded. The movies were grouped into 20 min time slots and tracking and diffusion analysis was performed (see Methods). Figure 2b shows the general experimental workflow for region assignment and tracking analysis. Fluorescence from the G3BP1$^{SNAP}$ constructs was used in a control channel to assign cellular regions for analysis. The cellular outline (white line) and the nuclear outline (blue line) were drawn manually (see Methods). The stress granules (red line) were assigned by an intensity threshold of the G3BP1$^{SNAP}$ signal and tracked dynamically over the whole movie. In each frame, single TDP-43$^{Halo}$ molecules were localized and their position was linked through successive frames of the movie, yielding single-molecule tracks (Fig. 2b). Tracks crossing from one region to another (e.g. from the cytoplasm to stress granules) were split at the region border and the parts of the tracks are assigned to the respective region (Fig. 2b).

### TDP-43$^{Halo}$ dynamic stress granule shuttling decreases with stress duration

In a first step, we assessed the stress-dependent shuttling of TDP-43$^{Halo}$ molecules in and out of stress granules (Fig. 2c–e). Figure 2c shows a significant decrease of the number of shuttling events with increasing stress duration. This effect is still visible when the number of shuttling events are normalized to the stress granule size (Fig. 2d). Such a decrease in the number of shuttling events can indicate either an increased interaction strength of TDP-43$^{Halo}$ inside of stress granules, or an overall decreased stress granule dynamics, potentially due to solidification of the granule[11,19]. Interestingly, the ratio between TDP-43$^{Halo}$ molecules entering or leaving the stress granules stayed constant over the whole stress period (Fig. 2e). Taken together, these observations can be explained by a decrease in stress granule dynamics with increasing stress duration and might hint towards a gain of solid-like properties in stress granules.

### Sodium arsenite stress reduces TDP-43 mobility

To investigate the mobility changes of TDP-43 under sodium arsenite stress, single-molecule tracking analysis was employed (Fig. 3). To biochemically validate the nature of TDP-43 slow-down, additionally, the solubility of TDP-43$^{Halo}$ and endogenous TDP-43 was assessed using a solubility assay and subsequent Western Blotting.

From the single molecule tracks, displacement histograms and cumulative displacement histograms are computed from all jumps within the tracks subjected to analysis (Fig. 3a). Diffusion coefficients and the respective fractions were computed by fitting the cumulative displacement histograms with a multi-exponential fit function[20,21] (see Methods). For the extraction of the apparent diffusion coefficients, a three-exponential fit function was used, since it fitted best (compared to mono- and double exponential) to the cumulative jump-distance histogram (Supplementary Fig. 3). This results in three diffusion regimes ($D_1$/slow, $D_2$/medium, $D_3$/fast). The three fractions ($F_1$/slow, $F_2$/medium, $F_3$/fast) indicate the proportion of TDP-43$^{Halo}$ molecules in each diffusion regime. An overview of the fitting parameters and fit quality is given in supplementary Fig. 3. Three diffusion coefficients allow to account not only for the assessment of a bound and mobile fraction but also for the aspect of anomalous diffusion[22,23]. To reduce the bias towards slower moving and bound molecules, only the first five jumps of every tracked single-molecule time-trace were used in the data analysis[23], however, similar results were obtained when all jumps were considered (Supplementary Fig. 4). For simplicity the effective diffusion coefficient $D_{eff}$ was computed (see Methods), representing the weighted average of all diffusion coefficients and respective fractions.

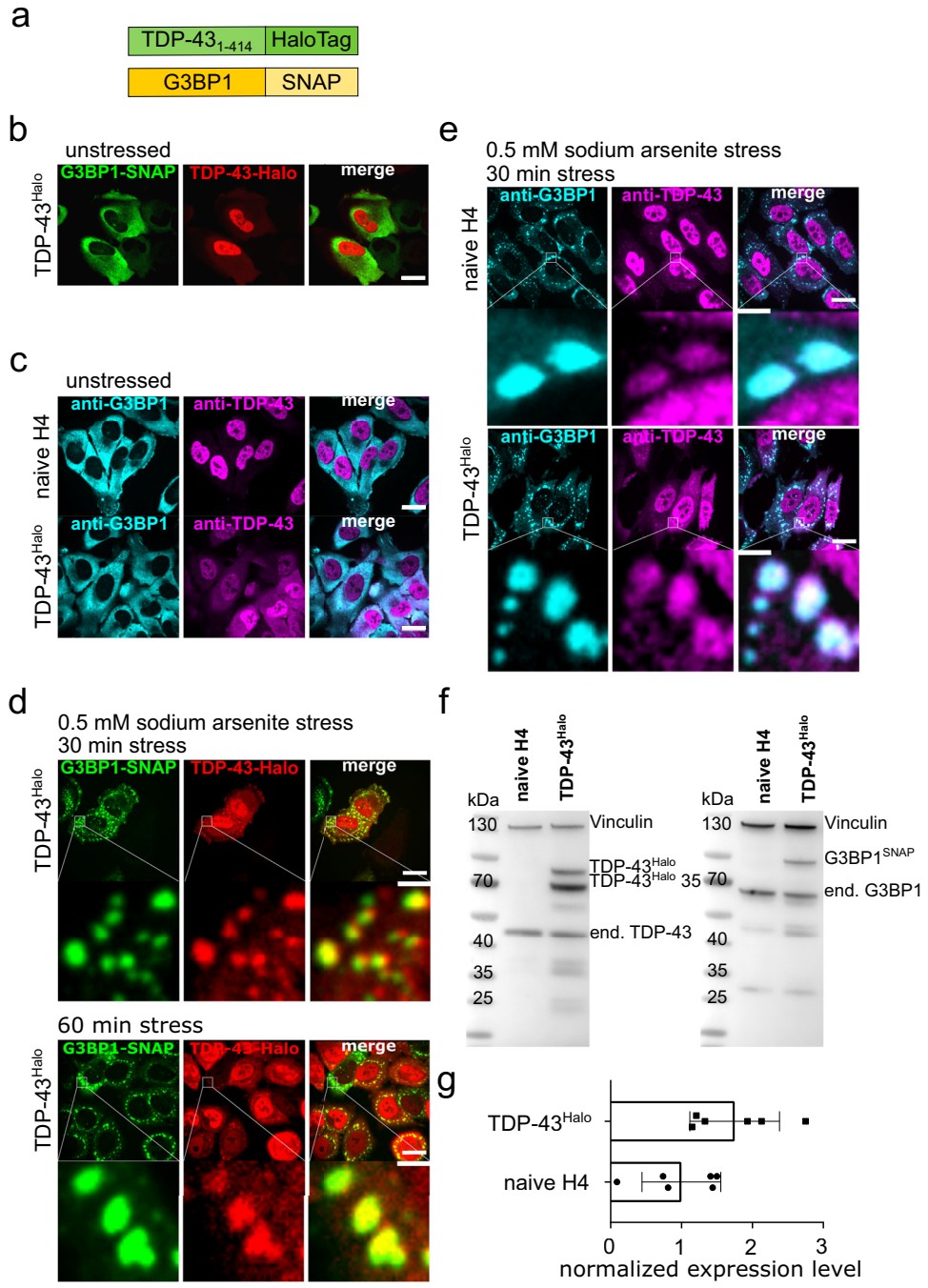

**Fig. 1 | Generation and quality control of TDP-43^Halo cell line. a** Schematic overview of the TDP-43^Halo and G3BP1^SNAP constructs. **b** Spinning disk confocal images of the TDP-43^Halo cell line under unstressed conditions (red: TDP-43-TMR, green: G3BP1-SiR, scale bar 20 μm). **c** Spinning disk confocal images of immuno-labeled naïve H4 cells as well as TDP-43^Halo cells under unstressed conditions (cyan: anti-G3BP1-Alexa532, magenta: anti-TDP-43-Alexa647, scale bar 20 μm). **d** Spinning disk confocal images of the TDP-43^Halo cell line under 30 min and 60 min sodium arsenite treatment (red: TDP-43-TMR, green: G3BP1-SiR, scale bar 20 μm and 2 μm). **e** Spinning disk confocal images of the immunostained TDP-43^Halo cell line and naïve H4 cells under 60 min sodium arsenite treatment (magenta: anti-TDP-43-Alexa647, cyan: anti-G3BP1-Alexa532, scale bar 20 μm and 2 μm). **f** Western Blot overview of the TDP-43^Halo cell line and naïve H4 cells stained with anti-vinculin, anti-TDP-43 or anti-G3BP1 antibodies showing proper expression of the transgenic constructs. The experiment was performed in triplicates. **g** Quantification of the overexpression for the TDP-43^Halo cell line compared to endogenous TDP-43 in naïve H4 cells shows a 1.5−2x overexpression of TDP-43^Halo, $n = 5$ independent preparations, data are displayed as the mean +/− STD. Source data are provided as a Source Data file.

Figure 3a depicts the data obtained from detected TDP-43^Halo tracks, recorded under unstressed and different stress conditions. In order to obtain a first quantitative comparison about TDP-43^Halo mobility as a function of stress duration, the effective diffusion coefficient $D_{eff}$ of TDP-43^Halo was determined for the whole cell (Fig. 3b). After 120 min of sodium arsenite stress, TDP-43^Halo showed a significant reduction in mobility as compared to the unstressed condition (unstressed: $D_{eff} = 4.94$ μm²/s, standard deviation: 0.84 μm²/s, stressed: $D_{eff} = 1.82$ μm²/s, standard deviation: 0.45 μm²/s, statistical test: Welch's $t$-test), suggesting oligomerization or aggregation of TDP-43^Halo or localization to small compartments that restrict mobility (Fig. 3b). As shown in Fig. 3c, an increase of the insoluble TDP-43^Halo fraction was observed with increasing stress, starting between 40 and 80 min after stress onset, although the band is rather faint as compared to the

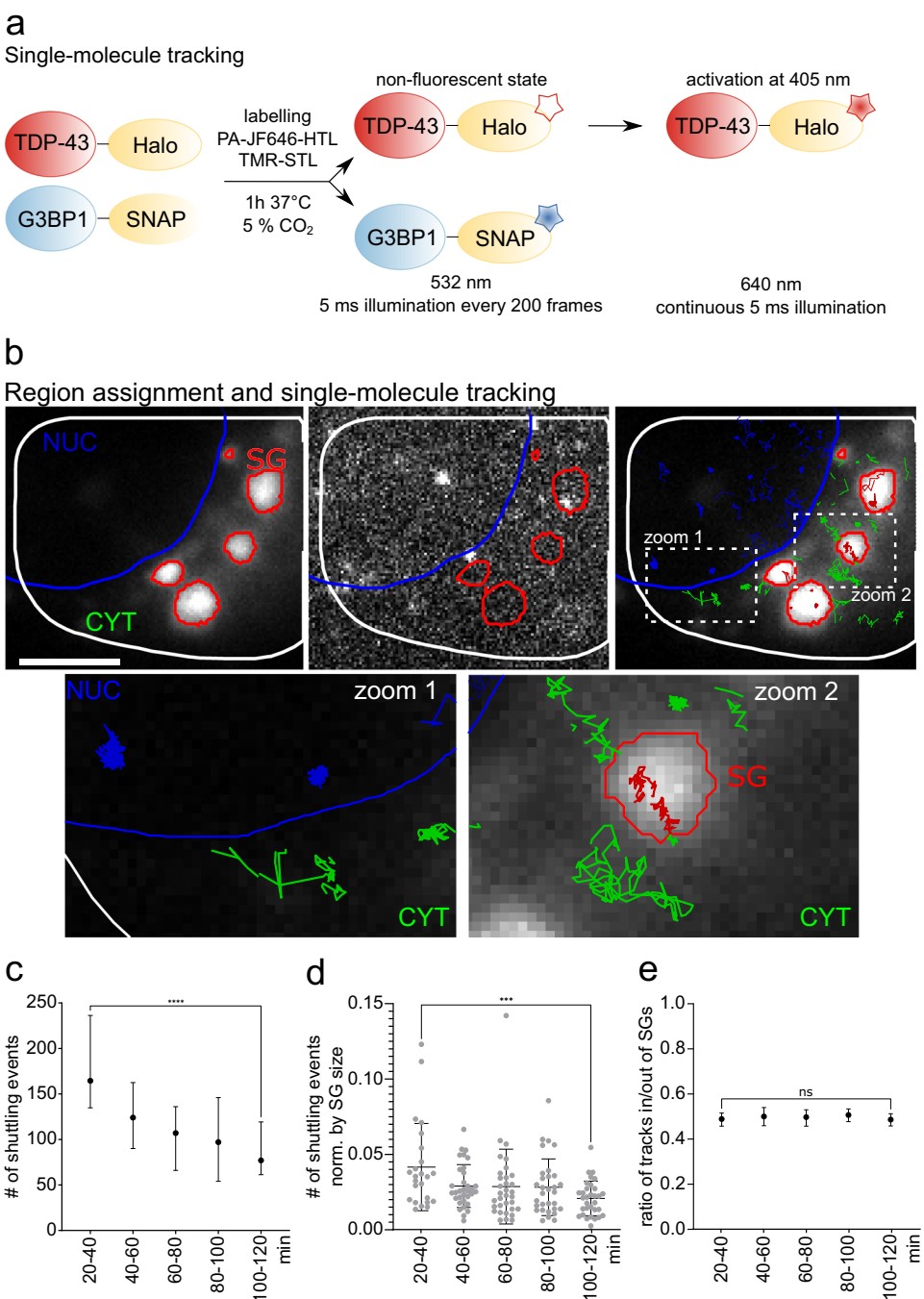

**Fig. 2 | Overview of the single molecule tracking experiments and diffusion analysis. a** Labeling and illumination scheme. TDP-43[Halo] and G3BP1[SNAP] are labeled with PA-JF646-HaloTag ligand (HTL) and TMR-SNAP-Tag ligand (STL) for 1 h at 37 °C and 5% $CO_2$ 24 h before imaging. TDP-43[Halo] is illuminated continuously with 640 nm and the PA-JF646-dye is continuously activated with 405 nm, the TMR-dye is illuminated with 532 nm and imaged every 200 frames. **b** Overview of the region assignment and region-specific track assignment (cytoplasm/tracking region: white outline and green tracks; nucleus: blue outline and tracks; stress granules: red outline and tracks). Zoom-ins and exemplary tracks for all regions are shown and show both, mobile TDP-43[Halo] as well as more immobile molecules. Scale bar 5 μm.

**c**–**e** Shuttling of TDP-43[Halo] in and out of stress granules. The **c** and **d**. Number of shuttling events significantly decreases with increasing stress duration (**c**: not normalized to SG size, values are displayed as median +/− interquartile range, p-value = < 0.0001, **d**: normalized to SG size, values are displayed as mean +/− STD, p-value = 0.0006). The ratio of TDP-43[Halo] tracks going in and out of stress granules stays constant over all stress durations (**e**), values are displayed as median +/− interquartile range. Statistical test: Two-tailed Mann–Whitney test. The number of analyzed cells per condition (n number) is given in Supplementary Table 1, the experiments cells were examined in independent experiments. Source data are provided as a Source Data file.

soluble fraction. Under unstressed conditions, however, all TDP-43 species were found in the soluble fraction. When comparing these data to the single molecule tracking data, it is likely that the reduced mobility of TDP-43[Halo] with increasing stress is partly caused by the formation of insoluble TDP-43 aggregates. The reduction of $D_{eff}$ is very pronounced, however, the bands depicting the insoluble TDP-43

fraction are rather faint. This indicates an aggregate-independent mechanism of TDP-43 mobility reduction, since the small amount of insoluble TDP-43 detected by the Western blot alone, cannot explain the observed strong reduction in TDP-43[Halo] mobility.

To get more detailed insight into the contribution of different TDP-43[Halo] species (fast, medium, slow) on the mobility reduction, the

**Fig. 3 | Sodium arsenite stress leads to a reduction of TDP-43$^{Halo}$ mobility observed in single-molecule tracking. a** Jump distance analysis and 3-exponential fitting results giving three different diffusion coefficients and fractions, slow ($D_1$/$F_1$), medium ($D_2$/$F_2$) and fast ($D_3$/$F_3$), in a stress-time-course experiment of TDP-43$^{Halo}$ cells. Stress duration increased from unstressed (magenta) to 120 min (green) of 0.5 mM sodium arsenite stress. Data are presented as mean values overlayed with the corresponding data points displaying the movie wise spread of data. **b** Stress time course of the effective diffusion coefficient $D_{eff}$ for the TDP-43$^{Halo}$ construct (whole cell, mean + STD, two-tailed Welch's $t$-test, $p$-value = <0.0001). Source data are provided as a Source Data file. **c.** Solubility assay of the TDP-43 construct. Solubility was assessed under unstressed conditions and different stress time-points. An increasing insoluble TDP-43 fraction was observed with increasing stress duration (unstressed, 40 min, 80 min and 120 min of 0.5 mM sodium arsenite treatment, anti-Vinculin, anti-TDP-43, $n$ = 3). **d.** Analysis of the different diffusion constants (slow/$D_1$/magenta, medium/$D_2$/light rose, fast/$D_3$/green) and the respective fraction within the stress time-course experiment (whole cell). For all experiments, the data are presented as mean values +/− STD and the standard deviations were calculated from the movie-wise distribution of the plotted value and statistical significance was assessed with a multiple unpaired t-test with Welch's correction. $P$-value ranges: <0.0001: ****, 0.0002: ***, 0.0021: **, 0.032: *, 0.123: ns. The number of analyzed cells per condition ($n$ number) is given in Supplementary Table 1, the experiments cells were examined in independent experiments.). Source data are provided as a Source Data file.

different diffusion states and the respective fractions were analyzed (Fig. 3d).

Figure 3d shows, that TDP-43$^{Halo}$ mobility reduction is caused by a decrease of the fast diffusion coefficient ($D_3$) accompanied also by a reduction of the fraction of fast TDP-43$^{Halo}$ species ($F_3$) resulting in an increase of both slow fractions, $F_2$ and $F_1$. Together, this suggests a general slow-down of TDP-43$^{Halo}$ with increasing stress duration, caused by mobility reduction and the formation of less-mobile TDP-43 species.

To get further biochemical insights into TDP-43 oligomerization and aggregation size-exclusion chromatography combined with TDP43 dot blotting was performed (Supplementary Fig. 5). After 120 min of sodium arsenite stress, a clear shift of TDP43 species eluting at earlier fractions (from 44 ml onwards) compared to late eluting TDP43 species (84 ml onwards) in the non-stressed condition was observed (Supplementary Fig. 5). This indicates higher oligomeric or aggregated species after 120 min of sodium arsenite stress and thus supporting the theory of TDP-43 oligomerization and aggregation with increasing sodium arsenite stress.

Several control measurements were conducted. As depicted in Supplementary Figs. 6a and 6b, no reduction of the effective diffusion $D_{eff}$ coefficient was observed for the C- and N-terminally tagged TDP-43 constructs during 120 min of measurement time under unstressed conditions. These results verify that the tracking environment or other

**Table 1 | Comparison of the effective diffusion coefficient $D_{eff}$ between TDP-43[Halo] and [Halo]TDP-43 constructs at different stress durations (unstressed, 60 min and 120 min 0.5 mM incubation with sodium arsenite, error given as the standard deviation (STD))**

|  | Whole cell | | | |
|---|---|---|---|---|
|  | TDP-43[Halo] | | [Halo]TDP-43 | |
|  | $D_{eff}$ [μm²/s] | STD [μm²/s] | $D_{eff}$ [μm²/s] | STD [μm²/s] |
| Unstressed | 4.940 | 0.838 | 2.683 | 0.741 |
| 60 min stress | 2.819 | 0.691 | 1.881 | 0.520 |
| 120 min stress | 1.819 | 0.452 | 1.036 | 0.263 |

external factors do not slow down TDP-43[Halo]. Also, the mobility of the HaloTag alone was assessed under stressed conditions. In this case, no mobility reduction was seen during 120 min of sodium arsenite stress (Supplementary Fig. 6c), thus excluding an unspecific stress-related mobility reduction.

Moreover, we also observed a similar effect of TDP-43 mobility reduction for the N-terminally tagged TDP-43 construct (Supplementary Fig. 7) ensuring that the observed decrease in TDP-43 mobility is independent of the HaloTag position.

Table 1 gives an overview of the effective diffusion coefficient $D_{eff}$ of the [Halo]TDP-43 and TDP-43[Halo] constructs at different time-points (unstressed, 60 and 120 min of sodium-arsenite stress, whole cell). The comparison shows, that [Halo]TDP-43 generally displays a lower mobility than the TDP-43[Halo] construct (as expected due to the lack of fragments in the case of [Halo]TDP-43) and that also for [Halo]TDP-43 a stress-related slow-down of mobility is observed.

In addition, we assessed TDP-43[Halo] and HaloTag mobility under 0.4 M D-Sorbitol, an oxidative and osmotic stressor (supplementary fig. 8)[24]. Sorbitol stress leads to an immediate, significant reduction of TDP-43[Halo] mobility in the whole cell and all cellular regions. Also for the HaloTag alone, we observed a decreased mobility throughout the whole cell. An immediate and comparable decrease of TDP-43[Halo] and HaloTag mobility under Sorbitol stress strongly argues for a general effect caused by the osmotic stressor. It was previously shown that sorbitol stress leads to cell shrinkage and an overall reduced mobility due to crowding effects[25,26], which is in agreement with our data.

Sodium arsenite is an oxidative stressor and was shown to damage mitochondria and cause ATP depletion[27]. Furthermore ATP depletion can lead to a loss in mobility as reported for several different proteins while other proteins seem to be unaffected[26,28–30]. For this reason, we assessed intracellular ATP-concentrations under unstressed and sodium arsenite stress conditions (60 and 120 min) and found an insignificant reduction of ATP levels after sodium arsenite stress (Supplementary Fig. 9), which is most likely insufficient to solely explain the observed strong reduction in TDP-43[Halo] mobility.

To investigate the role of ALS-causing TDP-43 mutations on TDP-43 mobility, we engineered TDP-43[Halo] constructs bearing either one mutation in the alpha-helical structure (M337V)[14,31] and another mutation in the Glycine-/Serine-rich domain (A382T)[14,32] (Supplementary Figs. 10, 11). Single-molecule tracking did not show any significant alterations in the course of mobility of mutant TDP-43 as compared to the wild-type constructs (Supplementary Figs. 12, 13), suggesting that the selected mutants do not have an additional effect on TDP-43 mobility with increasing stress. Before stress application and at low stress conditions we observed a faster mobility for familial mutants M337V and A382T (Supplementary Figs. 12, 13). Together, these results suggest that the pathological effect of the A382T and M337V mutations may not be based on an overall faster aggregation of mutated TDP-43 species.

## Longer stress leads to less efficient recovery of slow TDP-43 species

To find out whether and to which extent the reduced mobility of TDP-43[Halo] can be reversed, we again employed our single-molecule tracking analysis and bulk solubility assessment. Reversibility of TDP-43 slow-down was assessed by stressing TDP-43[Halo] cells for 60 min and 120 min with sodium arsenite and subsequent single-molecule tracking until 4 h after stress removal (Fig. 4a–f; statistical evaluation Supplementary Fig. 14). While short stress exposure (1 h) seems to allow almost complete restoration of TDP-43[Halo] mobility after 4 h of recovery, longer stress exposure (2 h) initiated processes hindering complete amelioration of TDP-43 mobility reduction (Fig. 4a, b and Supplementary Fig. 14a).

To get more insight, we again turned to the detailed analysis of the diffusion constants and respective fractions ($D_1/F_1$/slow, $D_2/F_2$/medium, $D_3/F_3$/fast) to test whether a reduced mobility is caused by a general slow-down of TDP-43[Halo] or a shift in the respective fractions. The fast diffusion coefficient $D_3$ was significantly reduced after 2 h of stress and both recovery conditions as compared to the unstressed condition (supplementary Fig. 14b). For the 1 h stress condition, $D_3$ was significantly decreased after 120 min of recovery. After 240 min of recovery, $D_3$ adapted comparable values for both stress conditions, although the decrease for the 1 h stress condition was no longer significant with respect to the unstressed condition.

The fast fraction $F_3$ was significantly reduced after 120 min of recovery (Supplementary Fig. 14b) and a longer stress duration led to a lower fast fraction than a shorter stress duration (1 h stress: $F_3 = 0.37$, 2 h stress: $F_3 = 0.24$). While 4 h after stress, $F_3$ showed complete recovery for cells stressed for 1 h, for longer stress duration $F_3$ remained reduced even after 4 h of recovery. This again underlines the interplay between two different effects that lead to a reduced TDP-43 mobility: (1) a decrease in mobility, and (2) an increase of immobile TDP-43[Halo] species.

To further biochemically characterize the nature of the reduced mobility after recovery, the previously described biochemical solubility assay was performed after either 1 h or 2 h of sodium arsenite stress and 2 h or 4 h of recovery, respectively (Fig. 4g). After longer stress exposure, TDP-43[Halo] showed a significant insoluble fraction, even after 4 h of recovery, while after short stress almost no insoluble TDP-43 was detected (Fig. 4g). The region-specific analysis of the recovery data is shown in supplementary fig. 15 and region-wise recovery is comparable to TDP-43[Halo] recovery observed in the whole cell. Note, after 1 h and 2 h of sodium arsenite stress, stress granules could be only assigned until 100 min and 120 min of recovery, respectively (Supplementary Fig. 15c, f), since they dissolved afterwards. Exemplary time-lapsed movies of G3BP1[SNAP] covering 120 min of sodium arsenite stress and covering 4 h of recovery are shown in supplementary movies 1 and 2, respectively.

Together, single-molecule tracking data and biochemical characterization indicate that TDP-43[Halo] is capable to recover from short stress insults while longer stress leads to persistent, insoluble TDP43 aggregates.

## TDP-43 mobility is reduced in stress granules, nucleus and cytoplasm

To study TDP-43[Halo] mobility in different subcellular compartments and to identify the contribution of different subcellular fractions of TDP-43[Halo] to the overall mobility reduction seen in the whole cell, diffusion data were analyzed in a region-specific manner (Fig. 5).

TDP-43[Halo] showed the highest mobility in the cytoplasm (Fig. 5a) under unstressed conditions with an effective diffusion coefficient of 5.40 μm²/s (standard deviation: 1.07 μm²/s) and sodium arsenite stress lead to a significant and continuous reduction of the effective diffusion coefficient to 2.23 μm²/s (standard deviation: 0.59 μm²/s) after 120 min of sodium arsenite stress. Since the HaloTag was attached to the C-terminus of TDP-43 and TDP-43 can be fragmented N-terminally[33],

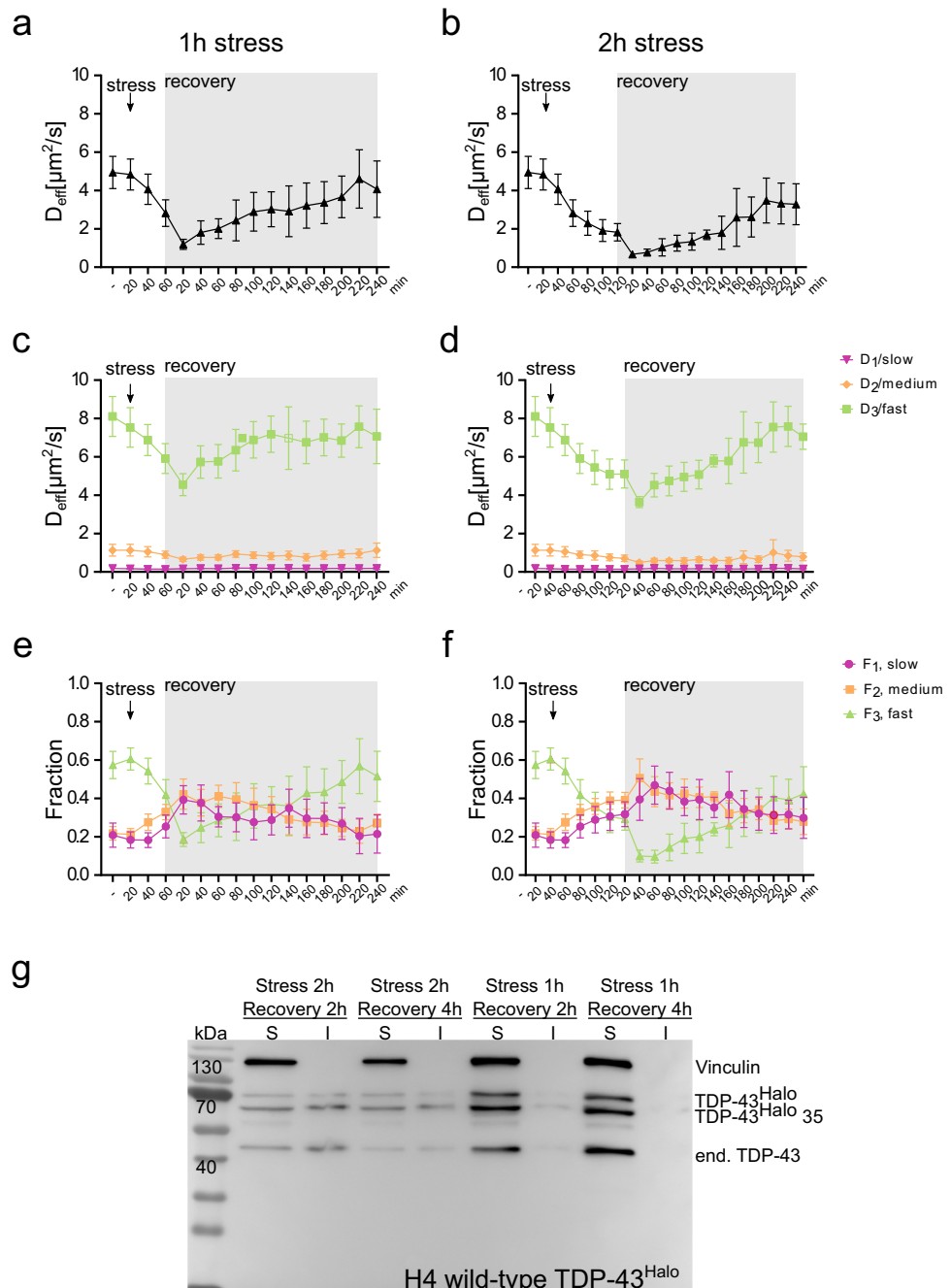

**Fig. 4 | Stress and recovery experiment. a**, **c**, **e** Stress and recovery time courses of the effective diffusion coefficient $D_{eff}$ and the diffusion coefficients $D_1$ (slow), $D_2$ (medium) and $D_3$ (fast) and their respective amplitudes ($F_1$, $F_2$, $F_3$) plotted for the whole cell for 1 h stress duration and up to 4 h of recovery (green: $D_3/F_3$, orange: $D_2/F_2$, red: $D_1/F_1$). Stress and recovery start points are marked by arrows. Source data are provided as a Source Data file. **b**, **d**, **f** Stress and recovery time courses of the effective diffusion coefficient $D_{eff}$ and the diffusion coefficients $D_1$ (slow), $D_2$ (medium) and $D_3$ (fast) and their respective amplitudes ($F_1$, $F_2$, $F_3$) plotted for the whole cell for 2 h stress duration and up to 4 h of recovery (green: $D_3/F_3$, orange: $D_2/$

$F_2$, red: $D_1/F_1$). Stress start points are marked by arrows and the recovery period is highlighted in gray. The number of analyzed cells per condition ($n$ number) is given in Supplementary Table 1, the experiments cells were examined in independent experiments. Source data are provided as a Source Data file. **g** Solubility assay of the TDP-43$^{Halo}$ wild-type after different stress and recovery durations (1: Stress 2 h, Recovery 4 h, 2: Stress 2 h, Recovery 2 h, 3: Stress 1 h, Recovery 4 h, 4: Stress 1 h, Recovery 1 h). Antibodies: anti-vinculin, anti-TDP-43, $n = 3$. For all experiments, the data are presented as mean values +/− STD and the standard deviations were calculated from the movie-wise distribution of the plotted value.

the construct visualizes both, full-length as well as fragmented TDP-43$^{Halo}$. A high mobility in the cytoplasm might therefore reflect the presence of full-length and fragmented TDP-43. Under unstressed conditions, TDP-43$^{Halo}$ mobility in the nucleus was in general slower as compared to the cytoplasm (Fig. 5b, $D_{eff} = 4.20$ µm²/s, standard deviation: 1.15 µm²/s), indicating an often bound or confined state of TDP-43 in the nucleus[11,34]. Notably, also in the nucleus $D_{eff}$ reduced

continuously with stress duration reaching the lowest value after 120 min with $D_{eff} = 1.26$ µm²/s (standard deviation: 0.50 µm²/s).

Stress granules are often discussed as the site of TDP-43 aggregation within the cell[8,35,36]. To test this hypothesis, we measured stress-dependent mobility of TDP-43$^{Halo}$ within stress granules (Fig. 5c). First stress granules could be assigned after 20 min of sodium arsenite stress. Although we observed a substantially lower TDP-43$^{Halo}$ mobility

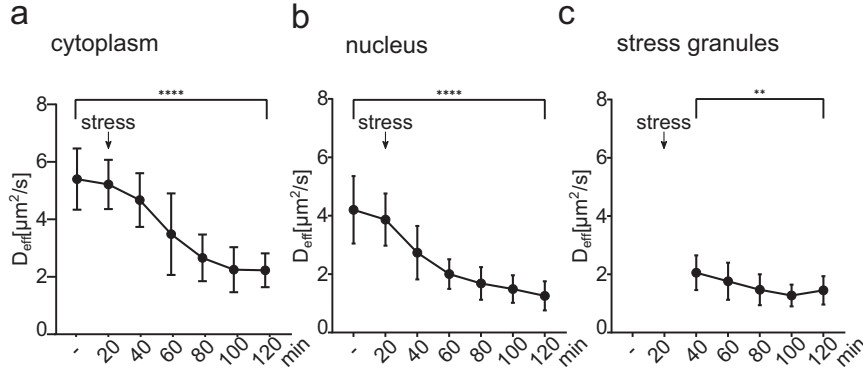

**Fig. 5 | Region- and diffusion class specific analysis of stress-induced TDP43 mobility reduction.** Stress time course of the effective diffusion coefficient $D_{eff}$ for the TDP-43[Halo] construct shown for the cytoplasm (**a**), the nucleus (**b**) and stress granules (**c**) (mean + STD, two-tailed Welch's *t*-test, cytoplasm *p*-value = <0.0001, nucleus *p*-value = <0.0001, stress granules *p*-value = 0.0010). The number of analyzed cells per condition (*n* number) is given in Supplementary Table 1, the experiments cells were examined in independent experiments. Source data are provided as a Source Data file.

in the stress granules first assigned at 20 min ($D_{eff}$ = 2.05 μm$^2$/s, standard deviation: 0.60 μm$^2$/s), compared to the unstressed value in the cytoplasm ($D_{eff}$ = 4.20 μm$^2$/s, standard deviation: 1.15 μm$^2$/s), prolonged stress only led to a modest further reduction of mobility within stress granules reaching at 120 min a value of $D_{eff}$ = 1.45 μm$^2$/s (standard deviation: 0.49 μm$^2$/s). In fact, $D_{eff}$ in the cytosol after 120 min was comparable to that in stress granules during early stress. Thus, the decrease in mobility in the cytoplasm and nucleus between 40 min and 120 min after stress induction was far more dramatic than observed in SGs.

Taken together, we found region-specific effects of stress-induced TDP-43[Halo] slow down within different cellular regions. Within stress granules TDP-43[Halo] showed, already after short stress durations, an expected, strong reduction of mobility as compared to TDP-43[Halo] in the cytoplasm under unstressed conditions, accompanied by a further, moderate decrease in mobility with prolonged stress. Surprisingly, we also found pronounced decreased TDP-43 mobility in the nucleus and the cytoplasm upon sodium arsenite stress, suggesting that aggregation or oligomerization also occurs outside of stress granules.

## Super-resolution imaging reveals inhomogeneous spatial distribution and mobility of TDP-43

Previous studies indicated, that stress granules are not necessarily homogenous phase-separated compartments but can exhibit regions of higher density termed 'core' that are surrounded by a less dense 'shell'[18,37]. Using super-resolution microscopy, it was reported that stress granule components like G3BP1 or poly(A)-RNA localize in a distinct substructure within stress granules[37,38]. Thus, we were interested if TDP-43 exhibits a similar substructure within stress granules at different stress time points.

To obtain such high-resolution sub-compartmental spatial information immunolabelled TDP-43 and G3BP1 were imaged using stimulated emission depletion (STED) microscopy under unstressed and different stress durations (30 min, 60 min and 120 min of 0.5 mM sodium arsenite) in naïve H4 cells (see Methods). TDP-43 shows an inhomogeneous distribution with denser regions (higher intensity) and less dense regions (lower intensity) within stress granules (Supplementary Fig. 16). This supports the idea of an inhomogeneous distribution of TDP-43 within G3BP1-positive stress granules. In contrast G3BP1 appears more homogenously distributed throughout the stress granules. This could in part be attributed to a more saturated fluorescence signal due to the high G3BP1 density within stress granules.

The results of the STED measurements which were obtained in fixed cells indicate an inhomogeneous distribution of TDP-43 within different cellular compartments and in particular stress granules. Next,

we wanted to determine whether such inhomogeneities were correlated with local mobility changes and therefore turned again to live-cell single-molecule imaging.

Single-molecule tracking analysis showed a slow-down of TDP-43[Halo] movement with increasing stress duration (Figs. 3 and 5). A reduced mobility can be caused by several processes, e.g., by localization to a confined compartment, by interaction with other protein complexes, by pathological aggregation or by physiological interactions. Also solubility assessment of TDP-43 under different stress condition (Fig. 3c), confirmed an insoluble fraction with increasing stress exposure. To study the spatial distribution of the different TDP-43 mobility regimes, we performed TALM analysis (tracking and localization microscopy)[18,39,40] and investigated local TDP-43[Halo] diffusivities (diffusion mapping, DM) as a means to obtain a super-resolved diffusivity map[20,41].

In TALM analysis, the fitted position of every detected spot is marked and thereby a super-resolved image can be created from the tracking data. A more frequent localization of TDP-43 at a given location, indicates binding to cellular structures or aggregation events. Frame-to-frame position jumps of localized molecules can be converted into average diffusion coefficients for each pixel in the image[20], indicating regions of local high or low mobility. The combination of these two methods allows for a correlation between binding hotspots or regions with increased TDP-43 localization and local mobility patterns. The visualization of such patterns can help to elucidate the origin of the observed TDP-43 slow-down in the cytoplasm and stress granules.

Figure 6a shows an exemplary G3BP1 image and the respective TDP-43[Halo] TALM and DM images recorded under unstressed and stressed conditions. The G3BP1 image was used to assign the cellular regions, in particular the location of stress granules (see Methods). To clearly separate stress granules from the stress-granule-free cytoplasm, a conservative approach in stress granule detection was chosen, resulting in a slightly overestimated stress granule area (see Methods). For the diffusion mapping (DM) images, higher mobility is depicted with a blue and lower mobility with a red color scheme (Fig. 6a). Note, in order to increase sensitivity, mobility greater than 6 μm$^2$/s is depicted in dark blue.

Under unstressed conditions, TDP-43[Halo] showed numerous binding hotspots or clusters in the nucleus and stress granules and regions of increased TDP-43 localization, termed localization patches, in the cytoplasm, that all correlated with local low TDP-43[Halo] mobility (red).

Binding hotspots and localization patches appeared as bright spots (or regions) within TALM images, originating from frequent TDP-

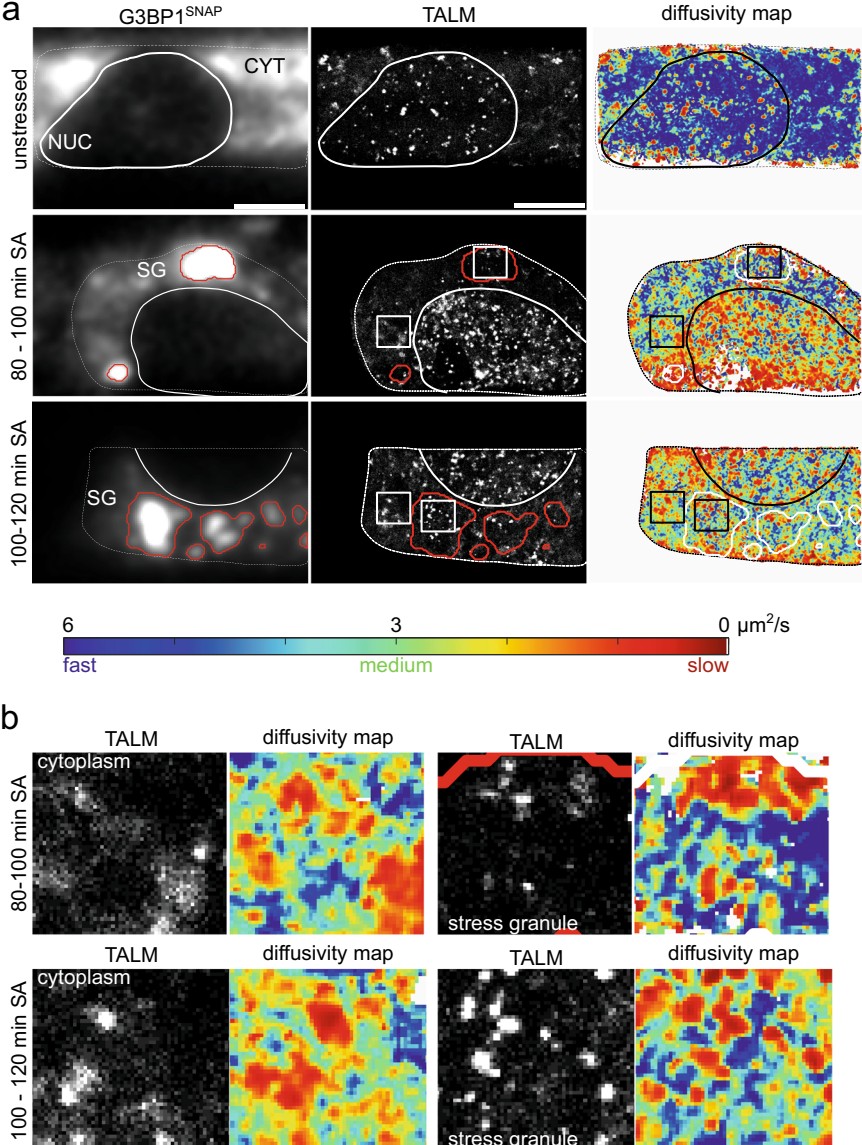

**Fig. 6 | TALM (tracking and localization microscopy) and displacement mapping (DM) analysis of TDP-43 movement in H4 cells. a** Overview of representative images obtained with TALM and DM analysis for TDP-43$^{Halo}$ imaged under unstressed and stressed (80–100 min and 100–120 min sodium arsenite treatment) conditions (color map: blue = fast movement, red = slow movement). Bottom: Color bar for the DM analysis. Blue color depicts faster movement and red color depicts slow movement. The maximum allowed jump distance for the analysis was 5 pixels. For better visualization, mobility >6 μm²/s is displayed in dark blue. **b** Crops of TALM and DM images of regions within the cytoplasm and stress granules. Local binding hotspots (TALM, white spots) correlate with a reduced TDP-43 mobility (DM, red regions) in both regions. An overview of the number of analyzed cells per condition giving similar results is given in Supplementary Table 1.

43 localizations from one, but more typical from several TDP-43 molecules (Supplementary Fig. 17).

Under unstressed conditions, diffusivity mapping clearly showed that despite the binding hotspots and localization patches, TDP-43 diffusion was strongly dominated by high TDP-43$^{Halo}$ mobility (blue dominated DM image) (Fig. 6a, upper panel). Such a mobility pattern of TDP-43 could be explained by the physiological shuttling between the nucleus and cytoplasm[34].

For longer stress duration (80–100 min, 100–120 min), diffusivity mapping showed a shift towards medium and low TDP-43$^{Halo}$ displacements throughout the whole cell. This fits well to the observed TDP-43$^{Halo}$ mobility reduction in all cellular compartments obtained with single-molecule tracking (Fig. 3). To get more detailed insight into the TDP-43$^{Halo}$ behavior in different regions, Fig. 6b shows exemplary cropped areas from the cytoplasm and stress granules. TDP-43$^{Halo}$ showed distinct binding hotspots within stress granules under both

stress conditions. These observations can be explained by localized 'binding hotspots or clusters' within stress granules as previously observed for G3BP1 and IMP1[18] and are consistent with the inhomogeneous stress granule structure observed with STED microscopy (supplementary fig. 16). The binding hotspots in stress granules correlate with a low TDP-43$^{Halo}$ mobility in DM (red spots, Fig. 6b). These binding hotspots were caused by the repeated localization of TDP-43 to these regions since an overlay between a TALM image and track start points from all molecules localized in one movie showed that several TDP-43$^{Halo}$ tracks are originating from these binding hotspots detected within stress granules (Supplementary Fig. 17a). The same is observed for the cytoplasm, however, here a more dispersed distribution of track start points can be seen. In addition, Supplementary Fig. 17b depicts a kymograph of a stress granule region, showing repeated binding of several TDP-43$^{Halo}$ molecules during the measurement period.

Based on the unexpected finding that also in the cytoplasm reduced mobility of TDP-43[Halo] was observed with increasing stress, the spatial distribution of TDP-43[Halo] in the cytoplasm was of special interest. Figure 6b shows an inhomogeneous distribution of TDP-43[Halo] localizations within the cytoplasm (TALM). Most notably, areas with frequent localizations correlate with an observed reduced TDP-43[Halo] mobility (DM). We refer to these areas as cytoplasmic patches which were assigned using a mobility threshold as described in the Methods part. We observed that the cytoplasmic area, covered by these patches, is significantly increasing with stress duration (Fig. 7a), indicating stress-related patch formation. Furthermore, we performed an anisotropy analysis of TDP-43[Halo] diffusivity in the cytoplasm, stress granules and cytoplasmic patches as described previously[17], to assess anomalous diffusion and potential trapping of TDP-43 in these regions (See Methods, Fig. 7b). TDP-43[Halo] showed an anisotropy value of around 1 inside the cytoplasm (excluding the patches) at unstressed and all stress conditions, indicating pure Brownian motion. Inside stress granules TDP-43[Halo] exhibited an anisotropy value of around 1.3, implying some confinement and trapping of TDP-43 inside these granules. Interestingly, TDP-43[Halo] showed the highest anisotropy values in a range between 1.7 and 2.8 inside the cytoplasmic patches, indicating strong anomalous diffusion and trapping of TDP-43[Halo]. Note, a similar trend was also found when analyzing regions of identical sizes (Supplementary Fig. 18), ruling out a that the observed differences in anisotropies are caused by the different sizes of the respective regions.

These differences in TDP-43 diffusion within stress granules, cytoplasmic patches and the rest of the cytoplasm also are evident in the computed angular plots of TDP-43 motion in the respective regions (Supplementary Fig. 19). Furthermore, diffusion analysis showed a very low effective diffusion coefficient $D_{eff}$ in the range of 0.49–0.34 μm²/s inside of the cytoplasmic patches (Fig. 7c) further indicating a strong confinement, oligomerization or aggregation inside these patches.

To further exclude unspecific stress effects, we assessed stress-related patch formation of the HaloTag alone (Supplementary Fig. 20). The HaloTag alone showed a similar patch-covered area as compared to TDP-43[Halo] under unstressed conditions. However, and most importantly, the patch-covered area did not increase with increasing stress duration for the HaloTag alone, excluding a general mechanism of stress-related patch formation.

In addition to the analysis of diffusion in cytoplasmic patches, we further characterized the TDP-43 localization hotspots or clusters observed in the TALM images (Fig. 6) using DBSCAN cluster analysis (see Methods). Figure 7d shows that the density of TDP-43 localization clusters found within stress granules was increasing with sodium arsenite stress, whereas the observed cluster density in the cytoplasm (including the cytoplasmic patches) was independent of stress. Moreover, the observed stress-induced cluster density in stress granules reached a level of around 0.76 cluster per μm², which is significantly higher than the cluster density observed in the cytoplasm (0.1 cluster per μm²). Importantly, the detections per cluster and the

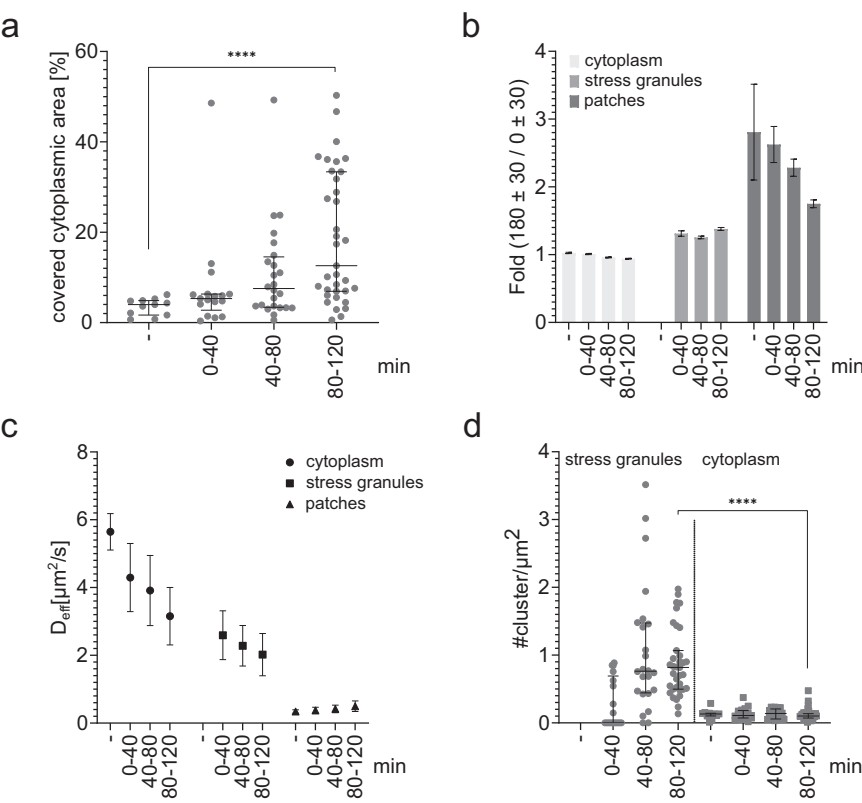

**Fig. 7 | Quantification of cytoplasmic TDP-43 patches and TDP-43 localization hotspots. a** Analysis of the cytoplasmic area covered by the patches at unstressed and different stress conditions, showing a significant increase of patch-covered cytoplasmic area with stress (stat. test: two-tailed Mann–Whitney, p-value = <0.0001, data are displayed as the median +/− the interquartile ranges). **b** Anisotropy analysis of TDP-43[Halo] molecules located in the cytoplasm (outside of cytoplasmic patches), stress granules and cytoplasmic patches at unstressed and different stress conditions. Values are displayed as the mean +/− the STD from 50 resamplings performed with 50% of the data. **c** Diffusivity analysis of TDP-43[Halo] molecules located in the cytoplasm (outside of cytoplasmic patches), in stress

granules and in cytoplasmic patches at unstressed and different stress conditions. The data are presented as mean values +/− STD and the standard deviations were calculated from the movie-wise distribution of diffusion coefficients. **d** Analysis of TDP-43[Halo] localization cluster density inside of stress granules and the cytoplasm (stat. test: two-tailed Mann–Whitney, p-value = <0.0001). Data are displayed as the median +/− the interquartile ranges). For all displayed data (**a**–**e**) the number of analyzed cells per condition (n number) is given in Supplementary Table 1, the experiments cells were examined in independent experiments. Source data are provided as a Source Data file.

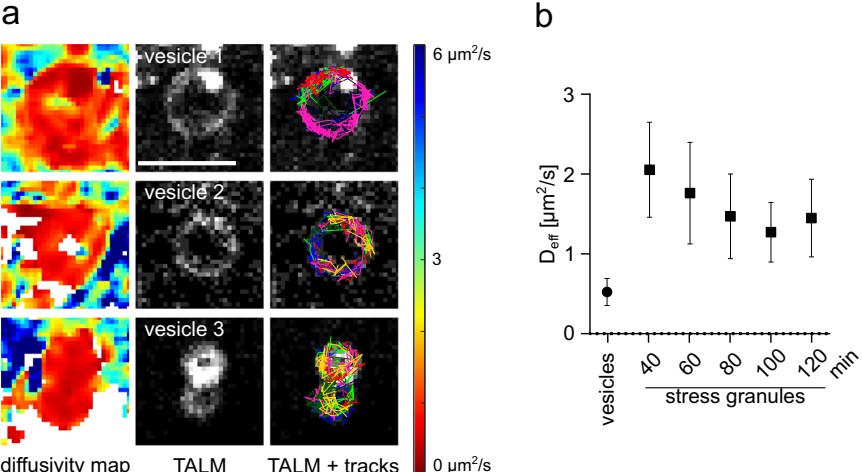

**Fig. 8 | Tracking and localization microscopy (TALM) and displacement mapping (DM) analysis of vesicular structures. a** DM (left) and TALM (middle) images showing exemplary vesicular structures observed within the cytoplasm. Selected TDP-43[Halo] tracks overlayed on top of the TALM image (right) show confined movement along the outline of the exemplary vesicles. Scale bar 1 μm, $n = 9$. TDP-43[Halo] mobility is displayed in a range of 0 μm²/s (red) to 6 μm²/s (blue). **b** Diffusivity analysis of TDP-43[Halo] molecules localized within the vesicular structure show a strongly reduced TDP-43[Halo] mobility as compared to the mobility observed within stress granules ($n = 9$ independent cells). Number of analyzed cells per condition for TDP-43 within stress granules is listed in Supplementary Table 1. For all experiments, the data are presented as mean values +/− STD and the standard deviations were calculated from the movie-wise distribution of the plotted value. Source data are provided as a Source Data file.

mean cluster area were not different between the cytoplasm and stress granules (Supplementary Fig. 21). Binding hotspots inside stress granules exhibited on average around 350 localizations per cluster. With an average track length of TDP-43[Halo] inside of stress granules of 10 frames, this corresponds to ~35 TDP-43 binding events per cluster. The mean cluster area was determined to be ~0.015 μm², corresponding to a 2D circle with a diameter of ~138 nm. This finding is consistent with observations made by Niewidok et al. determining a size of ~150–200 nm for stress granule binding regions for the stress granule proteins G3BP1 and IMP1[18]. Also Jain et al. determined the size of stress granule cores to be around 200 nm[37].

In addition, TALM and DM analysis showed TDP-43 localization accompanied by a reduced TDP-43[Halo] mobility to a vesicle-like compartment (Examples are shown in Fig. 8). Interestingly, TDP-43 was not equally distributed throughout the whole vesicle, but was localized more often at the outside than at the center of the vesicle and single TDP-43[Halo] tracks show TDP-43 movement along the vesicle outline, seemingly avoiding the interior (Fig. 8a). TDP-43[Halo] mobility was strongly decreased in the vesicles ($D_{eff} = 0.522$ μm²/s, standard deviation: 0.170 μm²/s), as compared to TDP-43 mobility inside of stress granules ($D_{eff} = 1.27–1.05$ μm²/s, Fig. 4c). One explanation for this behavior is the diffusion of a TDP-43 aggregates anchored to the vesicular membrane.

Taken together, TALM and DM analysis are valuable tools for the visualization and better interpretation of single-molecule tracking data. With these tools, we were able to show patches of increased TDP-43[Halo] localization in the cytoplasm that additionally correlate with a reduced mobility. The area covered by these patches increased with stress duration and showed a very low TDP-43[Halo] mobility. Diffusion of TDP-43[Halo] inside these patches is highly anomalous, indicating potential trapping of TDP-43[Halo] to these regions. Together, the results using the described methods suggest that TDP-43[Halo] is capable to oligomerize or aggregate in the cytoplasm independent of stress granules as a result of stress.

## Discussion

The presented study provides direct evidence that TDP-43 is capable to aggregate or oligomerize in the cytoplasm independent of stress granules. Using live cell single molecule tracking as well as spatial analysis of single-molecule tracking data we found numerous patches with reduced TDP-43 mobility in the cytoplasm, which we interpret as sites of TDP43 homo-oligomerization, aggregation or as sites of interaction with other cellular components.

Our generated stable, double-transgenic human H4 neuroglioma cell lines expressing Halo-tagged TDP-43 and SNAP-tagged G3BP1 serve as a model system that allow studying stress-related changes of TDP-43 mobility and localization on a single-molecule level. To our surprise, stress-induced TDP-43 mobility reduction was not limited to stress granules, but instead mobility in the nucleus as well as the cytoplasm were significantly reduced with increasing stress duration.

The mobility reduction of TDP-43 observed within stress granules was expected, due to the intrinsically high viscosity and high valency of interaction partners within stress granules. However, TDP-43 did also shows a moderate further decrease in mobility during prolonged stress inside of SGs, indicating aggregation of TDP-43 within stress granules or an overall loss of stress granule dynamics. Since distinct binding hotspots were already shown for other stress granule proteins like G3BP1 and IMP1[18] we wanted to investigate the sub-compartmental distribution of TDP-43 within stress granules. The application of tracking and localization microscopy (TALM) and diffusivity mapping (DM) analysis showed binding hotspots or clusters that correlate with reduced TDP-43[Halo] mobility. Further statistical and kymograph analysis of these localization hotspots confirmed the repeated binding of TDP-43[Halo] molecules to these regions and showed that the binding hotspots exhibit a diameter of ~138 nm, matching observations previously made by[18,37,38] for core-region sizes of different proteins localized within stress granules.

In addition, STED super-resolution microscopy showed that TDP-43 exhibits sub-structure within stress granules and becomes more homogenously distributed with increasing stress duration, which might be attributed to an increasing concentration of TDP-43 within stress granules at longer stress periods. A changing sub-compartmental distribution of TDP-43 and interaction with stress granules could elucidate the role of stress granules in the pathway of TDP-43 aggregation. Taken together, TDP-43 shows a sub-structural distribution within stress granules that might reflect possible binding sites observed using TALM imaging.

We also assessed TDP-43$^{Halo}$ dynamic stress granule shuttling and observed a decrease of all shuttling events with increasing stress duration. This can be interpreted by an increased TDP-43 binding or overall decreased stress granule dynamics with increasing stress duration. Maturation of phase-separated compartments, like stress granules, has been shown previously, and this could be an early step in the pathway of pathological protein aggregation in neurodegenerative diseases[10,11,42–47].

Next, we showed that the reduction of TDP-43 mobility is reversible after short stress durations. Longer stress durations lead to a reduced TDP-43 mobility after recovery, that can be attributed to a reduction in the diffusion coefficient and an increase of slower TDP-43 species. This finding is paralleled by the formation of insoluble TDP-43 species at longer stress durations seen by biochemical fractionation, suggesting that reversibility of TDP-43 aggregation and insolubility strongly depends on the stress duration applied. The impairment of protein degradation systems like autophagy or the ubiquitin proteasome system, may lead to TDP-43 aggregation or a defective aggregate clearance[48]. Whether impaired protein degradation is causative for the reduced recovery capability of TDP-43$^{Halo}$ was not part of this study and has to be investigated in future experiments.

Although a reduced mobility of TDP-43 within stress granules was expected, the strong reduction of TDP-43 mobility in the nucleus and especially the cytoplasm was surprising. 120 min of sodium arsenite stress, lead to a reduction of TDP-43 mobility of about 50% in the cytoplasm. Assuming no changes in the effective viscosity of the surrounding medium, such a decrease in diffusivity would correspond to a doubling of the hydrodynamic radius and thus to an eightfold increase of the effective mass and volume of the protein complex. These results might therefore suggest the formation of TDP-43 oligomers or smaller aggregates under the observed conditions.

Tracking and localization microscopy (TALM) and diffusivity mapping (DM) data further showed local patches of reduced TDP-43 mobility correlating with an increased TDP-43 localization whose formation and growth was dependent on the stress duration. The observed spatial distribution of TDP-43 within these cytoplasmic patches strongly differs from that observed in stress granules. Within stress granules, distinct binding hotspots or clusters were observed, whereas within the cytoplasmic patches, TDP-43 shows an almost homogeneous slow-down of TDP-43 molecules. TDP-43$^{Halo}$ mobility within these patches was highly reduced and anomalous, indicating potential trapping of TDP-43 in these regions. We therefore hypothesize that the TDP-43 slow-down observed in these patches throughout the cytoplasm might be a first step during oligomer and/or aggregate formation, thus giving rise to a stress granule independent pathway of TDP-43 aggregation or oligomerization.

These results are also in line with our previous work demonstrating TDP-43 oligomerization after sodium arsenite stress using a TDP43 protein complementation assay, leading to the formation of high molecular weight oligomers[49]. Using size-exclusion chromatography combined with dot blotting we confirm a shift of TDP-43 towards oligomeric and higher-order species after 120 min of sodium arsenite stress, similar to what had been seen previously for 60 min of sorbitol stress[49]. While these results nicely complement our data on reduced mobility under stress conditions, we still cannot fully exclude that the observed shift of TDP-43 species to larger complexes is due to interactions with other proteins or cellular components.

Another reason for the observed TDP-43 slowdown caused by sodium arsenite could have been reduced cellular ATP levels and mitochondrial impairments[27,50,51]. Moreover, recently also cytoplasmic TDP-43 aggregates and the formation of insoluble TDP-43 was shown after oxidative stress[52]. However, it was also shown, that ATP depletion leads to a general reduced mobility of some proteins, while others stay unaffected[26,28–30]. We assessed ATP levels at unstressed and different stress time-points and found an insignificant reduction of ATP levels in the TDP-43$^{Halo}$ cell line with increasing sodium arsenite stress duration. Thus, a reduction of ATP cannot be the cause for the observed slowdown. To further exclude a general effect, we assessed cytoplasmic patch formation for the HaloTag alone and were not able to observe a stress-related increase of cytoplasmic patches when looking at the mobility of the HaloTag alone.

Our data on TDP43 mobility reduction in the cytoplasm are in line with two recent studies demonstrating in other model systems TDP-43 aggregation or oligomerization in the cytoplasm. Using an optogenetic clustering system to induce controlled phase separation of TDP-43 wild-type and RNA binding deficient mutants, Mann et al. report, that exclusively cytoplasmic, RNA-binding deficient TDP-43 forms stress granule independent aggregates, that exhibit pathological features like hyper-phosphorylation and are devoid of RNA[10]. TDP-43 localized to these inclusions did exhibit a reduced recovery capability in FRAP measurements as compared to TDP-43 within stress granules, indicating aggregation.

Gasset-Rosa et al. employed an exclusively cytoplasmic Δ-NLS-TDP-43 construct and investigated TDP-43 phase separation after the application of fragmented TDP-43 or FUS particles as well as additional sodium arsenite stress[11]. They showed, that exogenously applied TDP-43 of FUS particles can induce aggregation of endogenous TDP-43 and that these aggregates mostly, do not co-localize with stress granule markers. In addition, they were able to show that after sodium arsenite treatment, Δ-NLS-TDP-43 first localizes to stress granules but forms stress granule-independent assemblies at later time points. Similar to what was seen in ref. 10, these assemblies did exhibit a reduced recovery in FRAP experiments as compared to stress granule localized TDP-43.

Although both studies used different stress models and TDP-43 constructs, they could independently show TDP-43 phase separation and aggregation in the cytoplasm, highlighting the importance of cytoplasmic TDP-43 oligomerization in ALS pathogenesis and research.

Moreover, there are additional, important surveys that showed that TDP-43 aggregation can occur independently of stress granule formation or localization[8,53]. Lastly[9], showed that TDP-43 stress granule localization is mediated by PAR-binding and that PAR-binding deficient mutants are no longer able to translocate to stress granules and subsequently form aggregates. All these studies attribute a rather protective role to TDP-43 stress granule localization and RNA interactions and highlight the importance of studying TDP-43 also outside of these compartments. This effect might not only be applicable for TDP-43, since[54] showed a similar, protective effect of stress granule localization for the RNA-binding protein FUS.

ALS pathology was shown to spread in patient's brains[55] and the search for a pathological, spreading species is a highly relevant topic. Previously we have shown evidence for cellular TDP-43 oligomerization by complementation essays and size exclusion chromatography[49] and we speculated that TDP-43 oligomers might serve as pathogenic species for propagation of pathology. In addition, TALM analysis revealed that TDP-43 can localize to confined, vesicle-like structure, where TDP-43$^{Halo}$ was localized exclusively to the outer layer of this structure and mobility of TDP-43$^{Halo}$ in these structures was highly reduced. Similar TDP-43 containing structures were already seen for nuclear TDP-43[56,57]. Since the localization of TDP-43$^{Halo}$ to such vesicular structures was rarely observed, it was beyond the scope of this work to identify the underlying nature of these vesicles. Potential target structures could be vesicles of the protein degrading machinery or late endosomes which might be released from the cell as exosomes[58–61]. In order to differentiate between these pathways further investigation is needed in the future.

The reduction of $D_{eff}$ of TDP-43 in the nucleus could be explained by an increased nucleic acid binding with increasing stress duration or a shift of the mobile TDP-43 fraction to the cytoplasm, which is a

commonly observed feature of TDP-43 under stressed conditions[34,62,63]. Tracking and localization microscopy (TALM) showed, that under stressed conditions, TDP-43[Halo] exhibits distinct and frequent binding hotspots within the nucleus. This argues for an increase in TDP-43 binding or aggregation within the nucleus with longer stress, which is in accordance with several studies reporting on TDP-43 phase-separation in the nucleus under unstressed and stressed conditions[11,56,57] and our data gives further evidence for these effects.

In conclusion, we were not only able to demonstrate a stress-related slow-down of TDP-43 mobility within stress granules, but also a reduction in TDP43 mobility in the cytoplasm independent of stress granules and in the nucleus. TDP-43 shows a stress-related and reversible decrease of mobility in all cellular compartments, indicating stress-related binding, oligomerization and aggregation events. The unexpected prominent and inhomogeneous reduction of TDP-43 mobility in patches of high TDP-43 concentration in the cytoplasm strongly argues for a probably aggregation-promoting dynamic of TDP-43 distinct from TDP-43 located to stress granules.

## Methods

### Cloning procedure
The phage UbiC G3BP1-SNAP plasmid was purchased from Addgene (Plasmid #119949) and characterized and published previously[64]. The TDP-43 constructs or the HaloTag only construct were cloned into an LVTetO lentiviral backbone[65]. An insert was cloned into the plasmid to introduce necessary restriction sites. The used primer and insert sequence are listed in Table 1. To generate the TDP-43 mutants M337V and A382T, single-point mutations were introduced to the respective wild-type plasmids by site-directed mutagenesis using the Q5 site-directed mutagenesis Kit (NEB). The used primers are listed in Table 2.

### Stable cell line generation
Lenti-X™ 293T cell line (Clonetech Labratories, # 632180) necessary for lentiviral particle production, were thawed one week before transfection. Lenti-X™ cells were grown to 80% confluence on a 10 cm dish (Sarstedt) prior to transfection. Cells were transfected using 10 μg transfer plasmid containing the respective TDP-43 construct, 7.5 μg packaging plasmid psPAX2 (Addgene plasmid #12260) and 2.5 μg envelope plasmid pMD2.G (Addgene plasmid #12259) using the jet-PRIME transfection system (Polyplus transfection). All three plasmids were mixed with 500 μl jetPRIME buffer and vortexed for 20 s. 30 μl jetPRIME transfection reagent was added and the mixture was again vortexed for 20 s. The mixed solution was incubated for 10 min at room-temperature and added dropwise to the Lenti-X™ cells. The transfected cells were incubated for 2 days at 37 °C and 5% CO₂ to ensure proper viral particle production.

One day before viral transduction the target H4 cells (catalogue #: ATCC HTB-148) were seeded on a six-well dish to reach 60% confluence the next day. At the day of viral transduction, the supernatant (9–10 ml) containing the viral particles was removed and 500 μl of the

supernatant was added to the H4 cells. For the generation of the double-transgenic cell lines 500 μl of viral particles containing the G3BP1-SNAP construct and 500 μl containing the respective TDP-43-Halo construct were added to one well respectively. To reach proper transduction efficiency the wells were incubated for 72 h at 37 °C and 5% CO₂. After 72 h the cells were once washed with DPBS, trypsinized and transferred onto a 10 cm dish. To remove cells without or with only one construct, the cells were sorted for double-positive clones at the Core Facility Cytometry at Ulm University. All used cell lines were checked for cross-contamination against the list of misidentified cell lines maintained by the International Cell Line Authentication Committee.

### Cell line cultivation
Sorted cell lines were cultivated in DMEM supplemented with 10% FBS, and 1% sodium-pyruvate, 1% non-essential amino-acids and 1% Gluta-max. The cell lines were kept in culture up to 10 passages. After that they were discarded and a new vial of frozen cells was taken into culture.

### Western blot sample preparation
**RIPA lysates.** For lysate preparation of unstressed cell lines RIPA buffer (sigma-aldrich, R0278) was used. Cells were seeded on a 10 cm dish and grown to 80–90 % confluence. For lysate preparation, cells were washed with PBS, trypsinized and transferred into a 15 ml falcon. The cell suspension was centrifuged for 5 min at 2000 × g (Centrifuge Beckman Coulter Allegra X-15-R, Rotor Sx4750A)) and 4 °C. The supernatant was removed and the pellet was washed with 5 ml ice-cold PBS. The suspension was again centrifuged for 5 min at 2000 × g and 4 °C. The supernatant was discarded. The pellet was dissolved in 400 μl ice-cold RIPA buffer and transferred into a 1.5 ml Eppendorf tube. The lysate was incubated for 30 min on a rotating wheel at 4 °C and centrifuged at 16,000 × g for 20 min at 4 °C (Eppendorf centrifuge 5417-R, Rotor F45-30-11). The supernatant was transferred to a fresh 1.5 ml Eppendorf tube and the pellet was discarded. The protein concentrations were adjusted to 1 mg/ml using a BCA assay (Thermo Fischer Scientific, 23227). Lysates were shock-frozen using liquid nitrogen and stored at −80 °C. Triplicates were prepared for each cell line.

For western blotting 24 μl protein lysate (1 mg/ml), 3.7 μl 1 M DTT and 9.3 μl 4× LDS were mixed, heated for 5 min at 95 °C and stored at −20 °C until usage.

**Solubility assay.** Cells were seeded on a 5 cm dish and grown to 80–90% confluence. Before the assay 5 ml fresh medium was added to the dish. The cells were either kept unstressed or treated 40 min, 80 min and 120 min with 0.5 mM sodium arsenite, respectively. After the end of the stress duration the medium was removed and the cells were washed with 2 ml PBS. The PBS was removed, 200 μl RIPA buffer was added per dish and cells were scraped of using a cell scraper. Lysed cells were transferred to a fresh 1.5 ml Eppendorf tube and sonicated

**Table 2 | Used primers**

| Name | Sequence | Purpose |
|---|---|---|
| Restriction-site-Insert | TTCTTAGAATTCTCTAGAAAATTCTCGAGGCAACTAGTGGCGCGCCAATGTA | Add. Xbal and Xhol site |
| TDP-43-Xbal-F | TTC GGC TCT AGA ATG TCT GAA TAT ATT CGG GTA AC | cloning |
| HaloTag-Ascl-R | TTA TTA GGC GCG CCA CTA GCC GGA AAT CTC GAG CGT AGA | cloning |
| TDP-43-Ascl-R | TTA AAT GGC GCG CCA CTA CAT TCC CCA GCC AGA AGA C | cloning |
| HaloTag-Xbal-F | GGC CGG TCT AGA ATG GCA GAA ATC GGT ACT GG | cloning |
| Q5_M337V_F | TTGGGGTATGGTGGGCATGTTAG | Site-directed mutagenesis |
| Q5_M337V_R | CTGCTCTGTAGTGCTGCC | Site-directed mutagenesis |
| Q5_A382T_F | TTCTGGTGCAACAATTGGTTG | Site-directed mutagenesis |
| Q5_A382T_R | TTAGAGCCACTATAAGAGTTATTTC | Site-directed mutagenesis |

three times for 3 s each. The protein concentration was determined by a Bradford assay and adjusted to 1 mg/ml for each sample using RIPA buffer. From each sample 100 μl lysate were transferred to a special tube for ultra-centrifugation (Beckman Coulter, 1.5 ml tube, 257448). The samples were centrifuged for 30 min at 100,000 × $g$ at 4 °C (Centrifuge Beckman Coulter Optima XPN-80, Rotor type 70.1). The supernatant, containing the RIPA-soluble fraction, was transferred into a fresh 1.5 ml Eppendorf tube. The pellet was washed with 200 μl RIPA, shortly sonicated and centrifuged for 30 min at 100,000 × $g$ at 4 °C (Centrifuge Beckman Coulter Optima XPN-80, Rotor type 70.1).). The supernatant was discarded and the pellet was dissolved in 100 μl 8 M Urea (8 M Urea, 20 mM Tris, pH 8). For better dissolution the sample was again sonicated three times for 3 s.

For western blotting 13 μl protein lysate (1 mg/ml), 2 μl 1 M DTT and 5 μl 4× LDS were mixed, heated for 5 min at 95 °C and stored at −20 °C until usage. Triplicates were prepared for each cell line.

**Western blotting**. Protein separation by size was done via SDS-PAGE. For the cell line expression overview ready-made gels (Invitrogen NuPAGE 4–12%. NP0335, NP0336) and buffers (Running buffer NP0002, Transfer Buffer NP0006-1) were used. For protein separation polyacrylamide gels with a 12% separating and a 5% stacking gel were prepared.

Samples were heated at 95 °C for 5 min before loading the onto the PAA-gel (marker: PageRuler Plus Prestained protein ladder (Thermo Scientific)). PAA-gels were pre-run for 30 min at 90 V. After that the voltage was increased to 125 V for 2 ½ h (SDS-running buffer: 25 mM Tris, 192 mM Glycin, 1 g/L SDS). The gel was transferred to the blot chamber (XCell II, EI9051) and blotted on a nitrocellulose membrane (Thermo Scientific, LC2008) for 3 h at room temperature (Blotting buffer: 25 mM Tris, 192 mM Glycin, 20% Ethanol). A proper protein transfer was confirmed by PonceauS staining and the membranes were blocked for 2 h with 5% skin-milk powder in 1× TBS-T (1× TBS, 0.05% Tween-20). Membranes were incubated overnight with primary antibodies (1:1000 in 5% skim-milk powder, anti-TDP-43 (rb, Proteintech 10782-2-AP), anti-G3BP1 (ms, sigma, WH0010146M1), anti-Vinculin (ms, Abcam, ab18058)) at 4 °C. Blots were washed three times for 20 min with 1× TBS-T to remove unbound primary antibodies. Blots were incubated with secondary antibodies for 1 h at room temperature (1:10,000 in 5% skim-milk powder, anti-rb-HRP-conjugate (Promgea, W401B)/anti-ms-HRP-conjugate (Promega, W402B)). After three times washes for 20 min with 1× TBS-T to remove unbound secondary antibodies and bands were visualized by adding ECL solution (Thermo Fischer Scientific, 32106) using the Fusion Solo7S Imager (Vilber). The Fusion software was used to analyze the overexpression of the TDP-43-Halo and G3BP1-SNAP constructs.

## Size exclusion chromatography and dot blotting

To prepare cell lysates for the size exclusion experiments, three 10 cm culture dishes were prepared per condition. The cells were either kept unstressed or stressed for 1 h or 2 h using 0.5 mM sodium arsenite. After the respective treatment, 400 μl PBS was added per dish, the cells were scraped off the dish and transferred into an Eppendorf tube. To induce cell lysis, the cell suspensions were sonicated three times for 3 s each. Protein concentrations were determined by the BCA assay (Thermo Fischer Scientific, 23227) and protein concentrations were adjusted to 1 mg/ml and a total volume of 3 ml. The stressed and unstressed cell lysates were used for size-exclusion chromatography (SEC) with the HiPrep 16/60 Sephacryl S-400 HR column (Cytiva) connected to an Äkta pure system (Cytiva). 2 ml of lysate were loaded onto the PBS equilibrated column and proteins were eluted with PBS at a flow rate of 0.5 ml/min. The protein eluate was monitored by UV absorbance at 280 nm and got separated into fractions of 1 ml for further experiments. Molecular mass of proteins was estimated using the Gel Filtration Calibration Kit (Cytiva).

A Dot blot apparatus (Whatman Schleicher & Schuell Minifold I Dot-Blot system, GE Healthcare) was used to test the eluates of SEC for TDP-43 (rb, Proteintech 10782-2-AP) signal. The system was set up as described by the manufacturer. To immobilize the proteins on a membrane, 200 μl of each SEC fraction were applied into the wells of the device and transferred to a nitrocellulose membrane (Amersham Protran 0.2, GE Healthcare) by vacuum. The staining and immunodetection was performed as previously described for Western blot membranes.

## Assay for determining cellular ATP concentrations

Cells were seeded on 5 cm culture dishes and grown for two days before the experiment. At the day of the experiment the cells were either kept unstressed or stressed for 1 h and 2 h with 0.5 mM sodium arsenite, respectively. After the respective stress duration, cells were washed with pre-warmed PBS and trypsinized. Cells were resuspended in pre-warmed DMEM and cell numbers were determined. For the determination of ATP levels, the CellTiter-Glo Luminescent cell viability assay (Promega, G7570) was used and the experiment was performed according to the manufacturer's recommendations. The experiment was performed in triplicates. For statistical evaluation a two-way anova was used.

## Spinning disk confocal microscopy

Cells were seeded on two-well Ibidi dishes (Ibidi GmbH, 80287) and labeled with 1 μM TMR-HaloTag ligand (Promega, G8251) and 0.12 μM SiR-SNAP-Tag ligand (NEB, S9102S) for 30 min at 37 °C and 5% $CO_2$. Cells were washed twice with PBS to remove unbound dye and stored in DMEM for at least 30 min prior to imaging. For imaging the medium was changed to OptiMEM. Cells were imaged at a custom-built spinning disk confocal microscope with temperature control and $CO_2$ supplementation. The microscope was built around an inverted microscope body (Axio Observer, Zeiss) together with a CSU10 scan head (Yokogawa). For imaging an oil immersion objective (UPlanSApo 60x/1.35 NA, Olympus) and for detection an EMCCD camera (D) were used. For excitation a 532 nm laser MGL-III-532-100 mW) and a 640 nm laser (Toptica iBeam-SMART 640-S) were used. Images were taken with a pixel size of 200 nm, an exposure time of 300 ms and averaging over 8 frames. Image analysis was performed with ImageJ. For the time-lapse movies of G3BP1-SNAP during stress and recoveries, cells were imaged under 5% $CO_2$ and 37 °C and images were taken every 30 s during the imaging period.

For a comparison between naïve H4 cells and the TDP-43 Halo cells were either kept unstressed or stressed for 60 min with 0.5 mM sodium arsenite. After the stress duration cells were fixed with 3.7% EM-grade PFA (Electron Microscopy Sciences, Catalog no. 15714) for 20 min at room-temperature. Samples were blocked with 3% BSA and 0.3% TritonX-100 overnight at 4 °C. Primary antibodies (1:500, anti-TDP-43 (rb, Proteintech 12892-1-AP), anti-G3BP1 (ms, sigma, WH0010146M1),) were incubated for 2 h at room temperature. Secondary antibodies (anti-rb-Alexa647 (Invitrogen A-21245) and anti-ms-Alexa532 (Invitrogen A11002), 1:500 each) were incubated overnight at 4 °C.

## Live-cell single-molecule imaging

H4 cells were seeded on a 35-mm Ibidi-dish with glass bottom (Ibidi GmbH, 81158) two days before imaging. One day before imaging cells were labeled with 10 nM photoactivatable-JaneliaFluor-646-HaloTag ligand (PA-JF646-HTL) and 0.3 μM TMR-SNAP-Tag ligand (TMR-STL) for 1 h at 37 °C and 5% $CO_2$. After three washing steps with PBS for 5 min to remove unbound dye, cells were stored in DMEM. The medium was then exchanged three times during the day for fresh DMEM.

Imaging was performed on a custom-built wide-field fluorescence microscope first described here[66]. A top stage incubator system with incubation chamber (H101-MCL-NANO-ZS200, Okolab s.r.l., Italy) was

used to enable temperature control and $CO_2$ supplementation. Measurements were conducted at 37 °C and 5% $CO_2$.

Illumination and fluorophore activation was done using 640 nm (Omikron LuxX® 638-300, 200 mW), 532 nm (Cobolt Samba, 150 mW) and 405 nm (Toptica iBEAM-SMART-405-S) laser lines. For single-molecule detection an illumination power of 2 mW (640 nm) and an activation power of ~1 μW (405 nm) were used. Imaging was performed in HILO-mode with a 100× oil immersion objective (Plan APO TIRF 100× NA 1.45 Oil, Nikon). Detection was performed using EMCCD AndorSolis cameras for each channel (iXon 897 Ultra and iXon 897, Andor Technology) and an effective pixel size of 130 nm. The Andor-Solis software (Version 4.31.30024.0) was used to record the tracking movies.

### Statistics of tracking data
Lists of all tracks and analyzed cell number for each experimental condition or analysis type is given in Supplementary Tables 1 and 2.

### Data acquisition
Single-molecule tracking measurements were performed with an exposure time of 5 ms (total frame cycle time 6.742 ms) and under continuous low 405 nm activation of the PA-JF-646-dye. Cells were either imaged under unstressed or stressed conditions (0.5 mM sodium arsenite or 0.4 M D-Sorbitol) for 2 h. Per movie 10,000 continuous frames were recorded in the tracking channel (TDP-43-PA-JF-646). Every 200 frames an image in the G3BP1-TMR channel was taken. For the TALM (tracking and localization) analysis imaging was performed for 30,000 frames and 405 nm activation during the frame transfer time between each frame was used. For recovery measurements the cells were stressed for 60 min or 120 min with 0.5 mM sodium-arsenite and then imaged for 0–120 min and 120–240 min after stress removal.

### Data analysis
**Tracking and classification of tracks into regions.** Spot detection, tracking and diffusion analysis was performed using TrackIt[20]. Spot detection was performed with a threshold factor of two. Tracking was performed using the 'Nearest neighbor' algorithm, with a tracking radius of five pixels, a minimum track length of three, one allowed gap frame to bridge detection gaps and a minimum track length of two frames before a gap frame. Cell outlines were used as tracking regions and additional sub-regions were drawn to define three region classes—nucleus, cytoplasm and stress granules. The cell outline and nuclear outlines were drawn manually based on the signal from the G3BP1-SNAP control channel. To account for signal fluctuations, a 'moving average' filter of two frames was applied. For TALM and DM data, additionally a Gaussian blurring of the G3BP1 channel of two pixels was applied.

Stress granules were outlined by an intensity-based-threshold defined by the G3BP1-SNAP control channel and were assigned dynamically over the whole movie time-course, to adapt for the mobility of stress granules. To better assess region-wise TDP-43 mobility, tracks that crossed region borders, were split at the border and assigned separately to the respective region.

**Diffusion analysis.** Diffusion coefficients and fractions were computed by fitting the cumulative distribution of displacements with a three-exponential Brownian diffusion model[20,21]. The bin size of the jump distance histogram was set to 1 nm. To prevent an overrepresentation of immobile molecules, only the first 5 jumps per track were considered while jumps over gap frames were discarded. A three component Brownian diffusion model was chosen based on the fit quality assessed by SSE and adjusted $R^2$-error (Fig. S3). Three-exponential fitting results in three diffusion coefficients $D_i$ and respective fractions $F_i$ - slow ($D_1/F_1$), medium ($D_2/F_2$) and fast ($D_3/F_3$). From these the effective diffusion coefficient $D_{eff}$ was calculated:

$$D_{eff} = F_1{}^*D_1 + F_2{}^*D_2 + F_3{}^*D_3 \tag{1}$$

Movies where the number of jumps was less than 30 were discarded. As an additional filtering step we used the 95% confidence interval of the fit to account for erroneous fitting. Movies were discarded if the error was five times higher than the median error calculated from all movies within a region and a specific stress condition. Mean values and standard deviations of the diffusion coefficients, fractions and effective diffusion coefficients were then calculated from the movie-wise values of the diffusion constant or fractions of the remaining movies.

**TALM and single-molecule diffusion mapping.** For the visualization of the TALM and DM data a scaling factor of two was applied, increasing the number of pixels by a factor of 4 and resulting in an effective pixel size of 65 nm. In TALM analysis, the fitted position of every detected spot is assigned to the pixel in which it appeared and the detection count of the respective pixel increased by one, creating a super-resolved image from the tracking data[40]. Higher count numbers (bright spots) in the TALM image indicate a more frequent localization of TDP-43 at the given spot, indicating binding to cellular structures or aggregation events. The DM analysis gives a pixel-wise average of the effective diffusion coefficient in the image[20], indicating local pattern with either high or low mobility (red: slow, blue: fast). To create a heat map of local diffusion coefficients, first, a three-dimensional histogram was created by binning the squared jump distances of all jumps between two consecutive frames of a track onto $43 \times 43$-$nm^2$ grids (up-scaled by a factor of three compared to the original image). Next, a map of average squared jump distances was created by calculating the average of all bin entries within a window of $3 \times 3$ bins around each bin. Finally, the diffusion coefficient map was created, where the diffusion coefficient $D_{x,y}$ of a pixel at the position $(x,y)$ was calculated using $D_{x,y} = \frac{X}{4 \cdot \Delta t}$, where $\Delta t$ is the frame cycle time of 6.742 ms, and X is the average of the squared displacements in the $3 \times 3$ window around $(x,y)$. The analysis was performed based on[41]. Stress granule outlines are slightly overestimated, to avoid the inclusion of signal, originating from TDP-43 within stress granules, to the cytoplasm.

**Segmentation and quantification of patches with slow diffusivity in diffusion maps.** Patches of slow diffusivity were defined as pixel areas in the diffusion map, where the diffusion coefficient was <2 $μm^2$ within a connected area comprising at least 50 pixels (4.7 $μm^2$). Subsequently, morphological closing was performed, using a disk with a radius of 3 pixels as structuring element. Pixels with a diffusivity ≥2 $μm^2$, that were fully enclosed by patches, were assigned to patches. Finally, patches were quantified in each movie by calculating the average patch size and the percentage of cytoplasm covered with patches. For the latter, the sum of the area of all patches within a movie was divided by the area of the cytoplasm, that was not covered by stress granules.

**Segmentation and quantification of TALM images.** TALM images were created by binning the detections (fit with sub-pixel precision) of all tracked molecules onto $43 \times 43$-$nm^2$ grids (up-scaled by a factor of three compared to the original image). The pixel values of a TALM image therefore corresponds to the amount of tracked detections within a given pixel. TALM images were segmented by an unsupervised clustering method using Voronoi tessellation and density-based spatial clustering of applications with noise (DBSCAN), which is freely available at: https://github.com/arian-arab/Voronoi_Clustering_MATLAB. First, only the Voronoi-cells with an area in the lower quartile were used. Next, Voronoi-cells were identified as clusters, if at least 80 cells were connected. Quantification of the clusters was performed

separately for stress granules and cytoplasmic areas outside of stress granules.

**Angular anisotropy.** The Angular anisotropy was calculated as previously described by[17]. Tracks were first classified into bound and free segments, using a hidden Markov model (HMM)[17,67]. Bound segments were removed and angles were calculated from free segments, where the jump distances making up the angle were >0.13 μm. Jumps involving gap frames were discarded. The fold-anisotropy $f_{180/0}$ was defined as the ratio between the amount of jump angles between $180° \pm 30°$ and $0° \pm 30°$.

**Shuttling between stress granules and cytoplasm.** For a quantification of the shuttling between stress granules and cytoplasm, only free track segments were considered. Therefore, tracks were classified into bound and free segments as described above. Tracks were classified as shuttling events between stress granule and cytoplasm according to the positions of their appearance and disappearance. Tracks starting outside of stress granules and ending inside of stress granules were classified as entering stress granules, whereas tracks starting inside of stress granules and ending in the cytoplasm were classified as leaving stress granules. Subsequently, the percentage of molecules entering stress granules was calculated with respect to all shuttling events. In addition, the number of total shuttling events was calculated and normalized by the area of stress granules.

**STED super-resolution microscopy.** Cells were seeded on 8-well Ibidi plates with glass-bottom (Ibidi GmbH, 80827) two days before fixation. Cells were either kept unstressed or stressed using 0.5 mM sodium arsenite. After the respective stress duration (30 min, 60 min or 90 min), cells were fixed using 3.7% EM-grade PFA for 20 min at room temperature and washed afterwards three times with PBS for 5 min. Cells were permeabilized and blocked for 2 h at room temperature (Blocking buffer: 3% BSA and 0.3% Triton-X-100 in PBS). After primary antibodies incubation overnight at 4 °C in 1:10 blocking buffer (anti-TDP-43 (rb, 1:500, Proteintech 12892-1-AP), anti-G3BP1 (ms, sigma, WH0010146M1, 1:1500,)), cells were washed three times for 10 min with PBS to remove unbound primary antibodies. Secondary antibodies were incubated for 1 h at room temperature (anti-rb/ms-Atto647N (sigma 40839-1ml-F/sigma 50185-1ml-F), anti-rb/ms-Atto594 (sigma 77671-1ml-F/sigma 76085-1ml-F), either 1:500 or 1:1500). Cells were then washed three times for 10 min with PBS to remove unbound secondary antibodies. Samples were stored at 4 °C until imaging. Just prior to imaging, cells were covered in 97% TDE.

STED images were taken using a custom-built dual-color three-dimensional STED microscope described in ref. [68]. For improved cross-talk reduction the emission filter in front of the 590 nm APD was changed to 623/24 BrightLine HC (AHF).

For sample illumination a randomly polarized super-continuum laser source (repetition rate 1 MHz) was split into the excitation wavelengths of 568 nm and 633 nm and their respective depletion wavelengths of 710 nm and 750 nm. A typical excitation beam power of ~0.8 μw and a depletion beam power of ~1.3 mW were used during image acquisition. Confocal images were taken with a pixel size of 50 nm, at a dwell time of 200 μs per pixel and peak photon number of 160–200 counts. STED images were taken with a pixel size of 20 nm, at a dwell time of 300 μs per pixel and a peak photon number of ~150 counts. Image analysis was done using ImageJ. For better visualization Gaussian blur with sigma = 1.5 was applied.

**Statistical information.** All spinning disk confocal imaging experiments and STED measurements were repeated at least twice with similar results.

A list of analyzed cell number for each experimental condition or analysis type is given in Supplementary Table 1, a list of all found tracks for each experimental condition or analysis type is given in Supplementary Table 2. For statistical comparison of the tracking control data sets a Brown-Forsythe and Welch ANOVA test was used. For the comparison of stress time-points in the single molecule tracking data a multiple, unpaired t-test with Welch correction was used. For the statistical evaluation of cytoplasmic patches and stress granule shuttling, a Mann–Whitney test was used. All statistical analysis was performed using GraphPad prism version 9.0.1.

### Reporting summary
Further information on research design is available in the Nature Research Reporting Summary linked to this article.

## Data availability
The raw single-molecule tracking data, extracted diffusion parameters and other data supporting the findings generated in this study have been deposited in the Dryad Digital Repository under accession code: [https://datadryad.org/stash/share/WMl6kJQhaXGefNk-JqUp6xiK3w7uGIPbLckAo4anM4Q]. Following DOI is assigned to the data: [https://doi.org/10.5061/dryad.02v6wwq50]. Further supporting data generated in this study are provided in the Supplementary Information file. All raw single particle tracking data are available in a TrackIt compatible Matlab file format. The TrackIt algorithm is freely available. Source Data are provided with the article. For material and correspondence please contact JM, KMD. Source data are provided with this paper.

## Code availability
A Matlab version of TrackIt is available at: [https://gitlab.com/GebhardtLab/TrackIt/]. The Voronoi tessellation and density-based spatial clustering of applications with noise (DBSCAN), which is freely available at: [https://github.com/arian-arab/Voronoi_Clustering_MATLAB].

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

## Acknowledgements

We thank the Core Facility FACS of Ulm University for their help with cell sorting, with special thanks to Dr. Simona Ursu and Dr. Sarah Warth. The photoactivatable Janelia Fluor 646 HaloTag ligand (PA-JF 646-HTL) was kindly provided by Luke Lavis (Janelia Research Campus, Howard Hughes Medical Institute, USA). Halo-SiR ligand was kindly provided by K. Johnsson (Max Planck Institute for Medical Research, Heidelberg, Germany). We thank Ramona Bück (Neurology, University Clinic, Ulm) for her support in the cell culture and bio laboratory. We thank Astrid Bellan-Koch for her help in cloning the TDP-43 mutant plasmids. This work was supported by the Deutsche Forschungsgemeinschaft (DFG) Emmy Noether Research Group DA 1657/2-1 (LS, KMD), the Deutsche Forschungsgemeinschaft (DFG) CRC 1279 (JM, LS) and DFG Research Grant GE 2631/3–1 to J.C.M.G.

## Author contributions

J.M., K.M.D., and L.S. conceived the study; J.M., K.M.D., and L.S. designed experimental approaches; L.S. designed and performed cloning, cell line generation and imaging experiments; TV optical setup work and maintenance; VB performed the SEC experiments; T.K. derived the tracking algorithm; L.S. and T.K., and J.C.M.G. performed data analysis; J.M., K.M.D., and L.S. wrote the manuscript; J.H.W., A.C.L., and J.C.M.G. provided intellectual input.

## Funding

## Competing interests

The authors declare no competing interests.
