## [Peer Review File · Nature Communications]

REVIEWER COMMENTS

Reviewer #1 (Remarks to the Author):

In this interesting paper from Streit et al., the authors utilize advanced single particle tracking and super-resolution microscopy to characterize the mobility and the aggregation of TDP-43, an important protein that has been shown to localize in aggregates in neurodegenerative diseases such as amyotrophic lateral sclerosis. Upon stress, TDP-43 is known to accumulate in stress granules (SG), but it is currently unknown, whether recruitment to SG is a protective or a promoting event towards pathological aggregation is currently debated.

The authors show that, upon stress TDP-43 molecules slow down, and such slow down is not limited to stress granules, but rather widespread in the nucleus and in the cytoplasm. Recovery from stress results in a recovery of TDP-43 mobility only when the stress is maintained for < 1hr.

Further the authors show that TDP-43 localization in stress granules is not homogeneous, that different TDP-43 molecules can revisit the same site on SG multiple times, and that TDP-43 can also localize and diffuse on the surface of unidentified vesicular structures.

Overall the paper is interesting and well written, and the methods have been carried out with rigor and with the necessary controls. As detailed below, I believe that it is maybe not surprising to observe a global slow-down of TDP in conditions leading to TDP-43 aggregation. Nevertheless, I think that the presented results are interesting because they hint to the possibility that aggregation of TDP-43 can occur at sites other than SGs, with potential implications for the mechanism underlying the spreading of ALS pathogenesis.

I also liked that the authors have limited their discussion to describe their data in the context of the available literature, without excessive speculations. Nevertheless, I think that some additional experiments could increase the impact of the presented results, as I will detail below, together with some minor comments on some technical details.

1. While I agree that the observation that TDP-43 displays globally reduced mobility upon stress is one of the most important results of the paper, this does not appear to me as surprising as the authors present it. As the authors mention in the discussion, sodium arsenite (the stress source used throughout the manuscript) causes ATP depletion, and it is expected that ATP depletion result in a decrease of mobility of many cellular factors undergoing transient interactions (e.g. transcription factors, PMID: 15189848, PMID: 15024032, RNA binding proteins PMID: 12473688, epigenetic factors: PMID: 34313222 etc). This

raises the question on whether other stress sources (maybe not associated to reduction of intracellular ATP levels) can induce a similar slow down on TDP-43, and whether other aggregating proteins involved in ALS (FUS?) show a similarly global slow-down upon stress induction. I understand that these experiments might fall beyond the scope of this paper, but they would strongly direct the interpretation of the presented data, in my opinion.

2. It is interesting that TDP mobility in stress granules is comparable to the slowest average mobility observed in the cytoplasm at late time points. (Figure 3C). Is at this time point a dynamic shuttling of TDP-43 in and out of SGs? Have the authors tried to analyze how frequently single TDP-43 molecules get trapped/escape stress granules at different timepoints. This might point to TDP acquiring more solid-like properties upon 'chronic' stress, also providing an explanation for their 'recovery' experiments.

3. Related to this, it was unclear to me whether the recovery experiment was performed on the whole cytoplasm (including SGs) or only in the cytosolic fraction. It would be interesting to see whether TDP-43 is more refractory to recovering when incorporated in SGs.

4. Talm analysis. For the insets in Fig 4B, it looks like the cytosolic patches in which TDP-43 show low mobility are larger than the hotspots observed in SG. The impression from these crops, is that while in SG TDP is highly mobile, with the exception of the binding hot-spots, in these cytosolic patches the TDP-43 diffusion is slowed down, and maybe rendered more anomalous, potentially trapping TDP in these regions. It would be very interesting to monitor the anomaly of diffusion in these patches, to check whether these zones trap TDP. One simple approach could be to analyze the anisotropy of displacements as in Hansen et al., 2020 PMID: 31792445).

5, The figure about the 'vesicle' is very interesting, but it raises two questions: a. How frequent are these observations. b. What is the substrate there? If these events are sufficiently frequent, maybe the authors could try to perturb possible candidates (lysosomes, large endosomes, others?), to try to figure it out.

Minor-points:

- In the discussion the authors speak about TDP-43 oligomerization as the cause for its slowdown. but I think that aggregation would be a better term. Indeed, the current data could be explained both by the formation of large (homo)oligomers, but also by interactions with scaffolds and non-soluble cytosolic components.

- The authors have chose to only account the first 5 displacements for their analysis of TDP diffusion. I am aware that this has been proposed before by Hansen, 2018 to avoid counting of the slowest molecules (that remain in focus for longer), but the choice of the max track length always seemed arbitrary to me, and it is easy to show that this can have the net effect of under-representing the immobile population. It's not a big deal because the authors mostly focus on relative differences, however it would be advisable to use other methods to correct for disappearance of molecules going out of focus (e.g. explicitly correcting the different populations for their probability of going out of focus).

- In Figure S.1 the IF controls Fields of view are the same as the ones used in Figure 1. It would be advisable to either choose different FOVs or specify in the figure legend that the same FOV were used - to avoid being potentially notified for 'duplications'.

- Numbering of the last two supplementary figures is off.

- The numbers on the x-axis of Fig 4 A, C, E are in the wrong order.

- Line 462 (third paragraph in the discussion). The sentence is incomplete. Maybe "during prolonged stress"? In SGs?

Reviewer #2 (Remarks to the Author):

In this manuscript, Streit et al. analyse the mobility and localization of TDP-43 under different stress conditions. For most of the experiments, they relied on a cell line moderately overexpressing Halo-tagged TDP-43. A major finding of the study is that the mobility of cytoplasmic TDP-43-Halo changes at stress conditions, which suggests that TDP-43 oligomerization occurs already in the cytoplasm independent of stress granules. This is an interesting finding that merits publication.

However, I do have concerns on the way the data are presented.

Throughout the manuscript, I am missing information on the size of the data set that has been analysed. For example, I did not see any information on the number of cells and single molecule tracks that have been analysed.

In the Reporting Summary the authors state that "the exact sample size for each experimental group/condition is given as a discrete number...". I did not find this information in the manuscript.

The authors also state in the same Reporting Summary “The number of cells subjected to tracking analysis was calculated (data available upon request)”. For me, this is a very unreasonable statement. Why is the number of cells analysed not given?

This information (the number of analysed cells; the number of tracks; experimental repeats) should be provided in the manuscript. Without such information, it is hardly possible to validate the reported experimental results.

Figures 5, 6, and 7 entirely lack a statistical analysis. This is not sufficient.

For example, Fig. 5 shows just one STED-image of a single cell for each condition. Presumably, these are representative images, but without an unbiased quantification, it is just not possible to draw any conclusion from these images.

There needs to be quantification of the observations reported in Figs. 5, 6 and 7.

Reviewer #3 (Remarks to the Author):

In the manuscript “Stress induced TDP-43 mobility loss independent of stress granules”, the authors generated transgenic lines expressing N- and C-terminally HALO-tagged TDP-43 to investigate the mobility of the tagged proteins under oxidative stress conditions in the cytoplasm, stress granules, and the nucleus, using live cell single-molecule tracking and STED super-resolution microscopy. The authors observed stress induced decrease of TDP-43 mobility in all compartments suggesting that TDP-43 oligomerization may take place in the cytoplasm separate from SGs.

A potential role of stress granules for the pathological phase transition of TDP-43 and other proteins containing intrinsically disordered regions undergoing LLPS is of interest and still not well understood.

However, the results are difficult to interpret and do not provide significant new insights into the pathophysiology of TDP-43 for the broad readership of this journal, as discussed in more detail below.

1) Adding more confusion than clarity, the authors interpret decreased mobility as changes in oligomerization in the context of pathology. (This is problematic by itself since other protein or RNA interactions may have a similar effect). However, homo-oligomerization of TDP-43 occurs under normal physiological conditions and is required for performing its RNA regulatory functions, such as mRNA splicing. This may be related but is different from pathological aggregation.

2) It is unclear whether the effect of sodium arsenite on protein mobility is specific for TDP-43, or to proteins undergoing LLPS, or a very general effect of oxidative stress. A shut-down of protein synthesis, ATP depletion, and many other fundamental metabolic changes in the cell (as mentioned in the Discussion section) may potentially cause widespread differences in protein mobility with very unclear significance for TDP-43 pathophysiology. For all we know this may affect GFP and other random proteins to the same extent.

3) The neuroglioma cell lines overexpress 33kDa HALO-tagged TDP-43 constructs that change the properties of the proteins. The C-terminal tag leads to a significant mislocalization of TDP-43 into the cytoplasm. This may be due to increased fragmentation and generation of truncated constructs missing the NLS and parts of the RRM domains. This physiological oligomerization of TDP-43 requires its N-terminal domain, and RNA binding requires the RRM1 & 2 domain. It is unclear from the data what specific or mixed protein species are being investigated in SGs and the cytoplasm. It is also unclear how the shift in mobility relates to pathology-relevant features such as RNA-binding, PTMs (phospho-TDP-43), fragmentation, and oligomerization/aggregation state of TDP-43.

4) The Ctrl lanes in the Suppl. Figure S4 western blot for HALO-TDP-43 show a prominent 35kDa fragment, while the soluble and insoluble fractions do not.

Reviewer #1:

The reviewer states that *“In this interesting paper from Streit et al., the authors utilize advanced single particle tracking and super-resolution microscopy to characterize the mobility and the aggregation of TDP-43, an important protein that has been shown to localize in aggregates in neurodegenerative diseases such as amyotrophic lateral sclerosis.”* and *“I think that the presented results are interesting because they hint to the possibility that aggregation of TDP-43 can occur at sites other than SGs, with potential implications for the mechanism underlying the spreading of ALS pathogenesis.”*

We appreciate that the reviewer is further positive about our work by stating *“Overall the paper is interesting and well written, and the methods have been carried out with rigor and with the necessary controls.”* Nevertheless, the reviewer thinks *“that some additional experiments could increase the impact of the presented results, as I will detail below, together with some minor comments on some technical details.”*

We are very pleased with the overall positive assessment of our paper by Reviewer #1. We are also grateful for the specific points raised and suggestions that helped us improve the quality of the manuscript.

Detailed response:

Point 1: While I agree that the observation that TDP-43 displays globally reduced mobility upon stress is one of the most important results of the paper, this does not appear to me as surprising as the authors present it. As the authors mention in the discussion, sodium arsenite (the stress source used throughout the manuscript) causes ATP depletion, and it is expected that ATP depletion result in a decrease of mobility of many cellular factors undergoing transient interactions (e.g. transcription factors, PMID: 15189848, PMID: 15024032, RNA binding proteins PMID: 12473688, epigenetic factors: PMID: 34313222 etc). This raises the question on whether other stress sources (maybe not associated to reduction of intracellular ATP levels) can induce a similar slow down on TDP-43, and whether other aggregating proteins involved in ALS (FUS?) show a similarly global slow-down upon stress induction. I

understand that these experiments might fall beyond the scope of this paper, but they would strongly direct the interpretation of the presented data, in my opinion.

Reply: We thank the reviewer for his/her helpful suggestion to study whether other stress sources, not associated to reduction of intracellular ATP levels, might induce a similar slow-down of TDP-43^{Halo}. Since sorbitol treatment has been shown to induce stress granule formation we applied sorbitol stress to our TDP-43^{Halo} cell culture model. As also observed using sodium arsenite stress, sorbitol treatment significantly reduces TDP-43^{Halo} mobility in the cytoplasm as well as in the nucleus. This effect could also be detected analyzing the whole cell. The new data are presented as a new supplemental figure S8. Sorbitol stress leads to an overall strong reduction of TDP-43^{Halo} mobility, already in the first 20 min of stress. A similar effect was obtained for the HaloTag alone, pointing to a general shrinkage of the cell caused by the osmotic stress that has been reported previously (Munder et al., 2016).

Additionally, since ATP depletion was referred to by the reviewer as a possible source for the observed mobility slowdown of TDP-43 in presence of sodium arsenite, we performed additional experiments to test how much the intracellular ATP levels are reduced given our experimental conditions. We observed an insignificant reduction of ATP levels, that is most likely not sufficient to explain the strong mobility reduction observed for TDP-43^{Halo} under sodium arsenite stress. Moreover, we also want to refer to our control experiments of HaloTag alone, that did not show a slowdown with sodium arsenite stress duration (compare supplementary figure S6c) nor an increase in the relative area covered by cytoplasmic patches with stress duration (supplementary figure S19), thus lending further support to the interpretation that the observed effect is specific to TDP-43. This finding is in accordance with previous studies demonstrating that other proteins do not show a reduction of mobility upon energy depletion e.g. (Munder et al., 2016), (Phair & Misteli, 2000), (Wagner, Chiosea, Ivshina, & Nickerson, 2004). These new data are incorporated in the revised version of the manuscript as new supplementary figure S9.

Furthermore, we agree with the reviewer that it would be highly interesting to study whether other aggregating proteins e.g. FUS would also show a general slow-down in the cytoplasm upon stress. However, we respectfully suggest that this might be subject of a future study since it falls beyond the scope of this paper.

Point 2: It is interesting that TDP mobility in stress granules is comparable to the slowest average mobility observed in the cytoplasm at late time points (Figure 3C). Is at this time point a dynamic shuttling of TDP-43 in and out of SGs? Have the authors tried to analyze how frequently single TDP-43 molecules get trapped/escape stress granules at different time points. This might point to TDP acquiring more solid-like properties upon 'chronic' stress, also providing an explanation for their 'recovery' experiments.

Reply: Following the reviewer's suggestion we determined shuttling events during sodium arsenite stress duration. We found that in the course of stress the number of shuttling events significantly decrease overall, but also when these events were normalized to the size of stress granules. Of note, when we analyzed the ratio of shuttling events in/out of stress granules, we observed a constant ration of around ~ 0.5. A detailed description and discussion of the new data is incorporated in the revised manuscript and the new data are presented as figure 2 c-e.

Point 3: Related to this, it was unclear to me whether the recovery experiment was performed on the whole cytoplasm (including SGs) or only in the cytosolic fraction. It would be interesting to see whether TDP-43 is more refractory to recovering when incorporated in SGs.

Reply: We apologize that we presented the recovery data for the whole cell including SGs (former Fig. 5) in a way that was obviously not clear to the reviewer. Following the reviewers' suggestion, we have now analyzed TDP-43^{Halo} recovery in a region-specific manner and determined recovery of TDP-43^{Halo} mobility in the nucleus, in the cytoplasm and stress granules separately. The region-wise recovery data are in accordance with the data obtained for the whole-cell, showing that after 2h of stress TDP-43^{Halo} mobility could not be fully regained after 4h of recovery. The new region-wise recovery data are displayed in supplementary figure S15.

Point 4: TALM analysis. For the insets in Fig 4B, it looks like the cytosolic patches in

which TDP-43 show low mobility are larger than the hotspots observed in SG. The impression from these crops, is that while in SG TDP is highly mobile, with the exception of the binding hot-spots, in these cytosolic patches the TDP-43 diffusion is slowed down, and maybe rendered more anomalous, potentially trapping TDP-in these regions. It would be very interesting to monitor the anomaly of diffusion in these patches, to check whether these zones trap TDP. One simple approach could be to analyze the anisotropy of displacements as in Hansen et al., 2020 PMID: 31792445).

Reply: We are grateful for the suggestion of the reviewer to analyze the anisotropy of displacements as described in the mentioned publication by Hansen et al. In the revised version of the manuscript we performed a detailed statistical analysis of the diffusion also probing for anisotropic displacements. Angular plots of TDP-43 located to the cytoplasm, stress granules and cytoplasmic patches under different stress conditions are shown as a new supplementary figure S18 and the results of the anisotropy analysis are shown in figure 7b. While a clear Brownian motion was detected inside the cytoplasm a strong bias of the angular distribution towards 180° of TDP-43 within cytoplasmic patches and stress granules became apparent. The anisotropy inside stress granules and the cytoplasmic patches was determined as ~ 1.3 and in a range of 2.8 – 1.8, respectively, indicating varying degrees of confinement in these two cytoplasmic compartments. A high anisotropy within the cytoplasmic patches hints towards highly anomalous TDP-43 mobility and a potential trapping of TDP-43 inside these patches. A description of the new data is incorporated in the revised manuscript (pages 22-23) and the new data are presented as new figure 7b and supplementary figure S18. Moreover, we performed further quantitative analyses of the cytoplasmic TDP-43 patches and the TALM images and the results are shown in the new figure 7 and in the reply to Reviewer #2.

Point 5: The figure about the 'vesicle' is very interesting, but it raises two questions: a. How frequent are these observations? b. What is the substrate there? If these events are sufficiently frequent, maybe the authors could try to perturb possible candidates (lysosomes, large endosomes, others?), to try to figure it out.

Reply: The observations of TDP-43^{Halo} localization to vesicle like structures was a rather rare event with about 1-2 observations per 10 to 15 cells. Additional exemplary

structures using tracking and localization microscopy (TALM) with matching diffusivity mapping images and exemplary tracks are demonstrated in the new figure 8 (adapted from former figure 7). Additionally, we analyzed TDP-43^{Halo} mobility within these structures and observed a mobility of $\sim 0.5 \mu\text{m}^2/\text{s}$ inside of these vesicle-like structures. Although the observation of these vesicle-like structures is highly interesting, their rarity makes it very hard to determine a substrate especially during live cell tracking experiments, when only the transgenic proteins can be visualized. Since co-staining with e.g. LysoTracker were not applicable due to incompatibilities in the available wavelengths, several cell lines co-expressing vesicular markers would be needed to determine the substrate in a live-cell imaging based approach. We kindly suggest that this might be subject of future studies investigating TDP43 mobility within vesicular like structures.

Minor-points:

Point 6: In the discussion the authors speak about TDP-43 oligomerization as the cause for its slowdown, but I think that aggregation would be a better term. Indeed, the current data could be explained both by the formation of large (homo)oligomers, but also by interactions with scaffolds and non-soluble cytosolic components.

Reply: We fully agree with the reviewer that we cannot distinguish between homo-oligomerization and/or the interaction/aggregation of TDP-43 with other components and we have clarified this in the revised discussion section. Moreover, to get further insights on the size of TDP-43 species under stress conditions we performed additional experiments, namely size-exclusion chromatography combined with TDP43-dot blotting. As shown in the new supplementary figure S5 application of 120 min of sodium arsenite stress clearly results in a shift of TDP43 from smaller species (fraction 84 - 110 ml, corresponding to \sim monomers) to bigger TDP-43 species (44 -74 ml factions corresponding to \sim >10-mers). These results nicely complement our data on reduced mobility under stress conditions, however, we still cannot fully exclude that the shift in TDP-43 species is due to interactions to other proteins. We acknowledged this aspect in the revised version of the manuscript.

Point 7: The authors have chosen to only account the first 5 displacements for their

analysis of TDP diffusion. I am aware that this has been proposed before by Hansen, 2018 to avoid counting of the slowest molecules (that remain in focus for longer), but the choice of the max track length always seemed arbitrary to me, and it is easy to show that this can have the net effect of under-representing the immobile population. It's not a big deal because the authors mostly focus on relative differences, however it would be advisable to use other methods to correct for disappearance of molecules going out of focus (e.g. explicitly correcting the different populations for their probability of going out of focus).

Reply: As suggested by the reviewer we analyzed TDP-43 mobility data of the whole cell either using the first 5 or all displacements. The comparison clearly shows that the consideration of all displacements leads to an overall reduced mobility due to the stronger bias towards bound or slowly moving TDP-43^{Hal} species. Nevertheless, the observed effect of an overall stress-induced reduction of TDP-43 did not change. This additional analysis is incorporated in the revised version of the manuscript as supplemental Figure S4.

Point 8: In Figure S.1 the IF controls Fields of view are the same as the ones used in Figure 1. It would be advisable to either choose different FOVs or specify in the figure legend that the same FOV were used - to avoid being potentially notified for 'duplications'.

Reply: We appreciate this comment, followed the suggestion of the reviewer and stated in the figure legend of supplementary figure S1 that we used the same FOV. Since the image was taken from naïve H4 cells neither expressing C-or N-terminally tagged TDP-43 it serves as a control for both C-and N-terminally tagged TDP-43.

Point 9: Numbering of the last two supplementary figures is off.

Reply: We thank the reviewers for his/her attention and corrected this error in the revised manuscript.

Point 10: The numbers on the x-axis of Fig 4 A, C, E are in the wrong order.

Reply: We kindly appreciate this comment and corrected it in the revised manuscript.

Point 11: Line 462 (third paragraph in the discussion). The sentence is incomplete. Maybe "during prolonged stress"? In SGs?

Reply: We apologize for this oversight and corrected it accordingly "prolonged stress in SGs".

Reviewer #2:

Also, reviewer#2 is positive about our work "This is an interesting finding that merits publication."

However, the reviewer also has "concerns on the way the data are presented".

Point 1: Throughout the manuscript, I am missing information on the size of the data set that has been analyzed. For example, I did not see any information on the number of cells and single molecule tracks that have been analyzed.

Reply: We sincerely apologize for not providing this information. Lists with all analyzed cells per experimental condition and all found tracks are now given in the supplementary tables S1 and S2.

Point 2: In the Reporting Summary the authors state that "the exact sample size for each experimental group/condition is given as a discrete number...". I did not find this information in the manuscript. The authors also state in the same Reporting Summary "The number of cells subjected to tracking analysis was calculated (data available upon request)". For me, this is a very unreasonable statement. Why is the number of cells

analyzed not given? This information (the number of analyzed cells; the number of tracks; experimental repeats) should be provided in the manuscript. Without such information, it is hardly possible to validate the reported experimental results.

Reply: We deeply apologize for this oversight. A list with all analyzed cells per condition is given in supplementary table S1. We also included the number of analyzed tracks in the supplementary table S2.

Point 2: Figures 5, 6, and 7 entirely lack a statistical analysis. This is not sufficient. For example, Fig. 5 shows just one STED-image of a single cell for each condition. Presumably, these are representative images, but without an unbiased quantification, it is just not possible to draw any conclusion from these images. There needs to be quantification of the observations reported in Figs. 5, 6 and 7.

Reply: As suggested by the reviewer we performed an unbiased quantification of the TALM and DM data. For the quantitative analysis of the TALM and DM images (figure 6) we observed an increase of cytoplasmic patches with increasing stress duration (figure 7). For the HaloTag alone (supplementary figure S19) no prominent patch formation and a stress related increase in the patch covered area was observed, excluding a general stress related effect.

As already mentioned in our reply to Reviewer #1, we further analyzed TDP-43 anisotropy within these patches (as performed in Hansen et al., 2020 PMID: 31792445) and observed a strong anisotropy of TDP-43^{Halo} within these patches, indicating trapping and anomalous diffusion of TDP-43^{Halo} in these regions. We further looked at the diffusion coefficient of TDP-43^{Halo} within these cytoplasmic patches and observed a diffusivity of $\sim 0.2 \mu\text{m}^2/\text{s}$ (Figure 7 b, c).

Additionally, we analyzed and quantified TDP-43 binding hotspots or cluster as observed in stress granules and the cytoplasm. We observed an increasing cluster density within stress granules with increasing sodium arsenite stress, indicating a stress related clustering of TDP-43^{Halo} within stress granules (Figure 7d). In the cytoplasm, we observed a constant and low cluster density, indicating that cytoplasmic cluster of TDP-43^{Halo} are forming in a non-stress-related manner (Figure 7d). Clusters

in stress granules and in the cytoplasm showed between 300 and 400 TDP-43^{Halo} detections (Supplementary figure S20a). With a mean track length of TDP-43^{Halo} tracks within stress granules of around 12, this corresponds to 25 to 33 binding events per clusters. There were no differences between the mean cluster size in the stress granules and the cytoplasm (Supplementary figure S20b). A mean cluster size of 0.02 μm^2 would correspond to a circle with a diameter of approx. 160 nm. Size dimensions of binding regions within stress granules fit to similar observations made by (Jain et al., 2016; Niewidok et al., 2018).

Since TALM and DM analysis gave a very in depth and thorough overview of the stress effect of TDP43 mobility we felt that STED imaging (former figure 5, which was recorded on fixed cells) would only give complementary information and therefore moved the STED imaging figure to the supplementary material part of the revised manuscript. Nevertheless, we provide additional representative images of stress granules as illustrative presentation of TDP- 43 and G3BP1 within stress granules (please see new supplemental figure S16).

In former figure 7 we presented an exemplary picture of TDP-43^{Halo} located to a vesicular structure as already stated in the response to point 5 of Reviewer #1, TDP-43^{Halo} localization to vesicle like structures was a rather rare event with about 1-2 observations per 10 to 15 cells. Additional exemplary structures using tracking and localization microscopy (TALM) with matching displacement mapping images and exemplary tracks are now presented in the new figure 8a. Additionally, we performed a statistical analysis of TDP-43^{Halo} movement within these structures (presented in figure 8b) and observed a mobility of $\sim 0.5 \mu\text{m}^2/\text{s}$ inside of these vesicle-like structures.

Reviewer #3:

The reviewer states “A potential role of stress granules for the pathological phase transition of TDP-43 and other proteins containing intrinsically disordered regions undergoing LLPS is of interest and still not well understood. “But the reviewer also states “the results are difficult to interpret and do not provide significant new insights into the pathophysiology of TDP-43 for the broad readership of this journal, as discussed in more detail below. “

We are very grateful, that the reviewer agrees with us regarding the scientific interest into the general topic of the manuscript, and we are confident, that, given the additional experiments and data presented in the revised manuscript the importance of the novel data obtained becomes more evident.

Point 1: Adding more confusion than clarity, the authors interpret decreased mobility as changes in oligomerization in the context of pathology. (This is problematic by itself since other protein or RNA interactions may have a similar effect). However, homo-oligomerization of TDP-43 occurs under normal physiological conditions and is required for performing its RNA regulatory functions, such as mRNA splicing. This may be related but is different from pathological aggregation.

Reply: The reviewer is of course right that homo-oligomerization of TDP-43 occurs under normal physiological conditions and is not necessarily related to pathological aggregation of TDP-43. As also stated in the response of point 6 of reviewer #1 we fully agree with the reviewer #3 that we cannot distinguish between homo-oligomerization and/or the interaction/aggregation of TDP-43 with other components and we have clarified this point in the revised manuscript. However, to get further insights on the size of TDP-43 species under stress conditions we performed size-exclusion chromatography combined with TDP-43-dot blotting. As demonstrated as new supplemental figure S5 application of 120 min of sodium arsenite stress clearly results in a shift of TDP-43 from species eluting at fraction 84-110 ml corresponding to ~monomers to TDP-43 species eluting at 44-74 ml fractions corresponding to >~10-mers. These results nicely complement our data on reduced mobility under stress conditions, however, we still cannot fully exclude that that the shift in TDP-43 species is due to interactions to other proteins or cellular components. We acknowledged this aspect in the revised version of the manuscript.

Furthermore, with the TDP-43^{Halo} cell line, we were able to track full-length, as well as fragmented TDP-43. The fragmented TDP-43 lacks the N-terminal domain required for dimerization. We therefore speculate that the observed mobility reduction might not solely caused by a functional dimerization of TDP-43.

Point 2: It is unclear whether the effect of sodium arsenite on protein mobility is specific for TDP-43, or to proteins undergoing LLPS, or a very general effect of oxidative stress.

A shut-down of protein synthesis, ATP depletion, and many other fundamental metabolic changes in the cell (as mentioned in the Discussion section) may potentially cause widespread differences in protein mobility with very unclear significance for TDP-43 pathophysiology. For all we know this may affect GFP and other random proteins to the same extent.

Reply: We thank the reviewer for his/her helpful suggestion to study whether other stress sources not associated to reduction of intracellular ATP levels might induce a similar slow-down of TDP-43^{Halo}. Since sorbitol treatment has been shown to induce stress granule formation we applied sorbitol stress to our TDP-43^{Halo} cell culture model. As also observed for sodium arsenite stress, sorbitol treatment significantly reduces TDP-43^{Halo} mobility in the cytoplasm as well as in the nucleus. This effect could also be detected analyzing the whole cell. The new data are presented as a new supplemental figure S8. Sorbitol stress leads to an overall strong reduction of TDP-43^{Halo} mobility, already in the first 20 min of Sorbitol stress. A similar effect was obtained for the HaloTag alone, indicating that the observed effect could be caused by a general the shrinkage of the cell due to osmotic stress (Munder et al., 2016).

In addition, given the reviewer's suggestion, we assessed the mobility of the HaloTag alone during sodium arsenite stress and did not observe a further reduction of mobility between 0-20 min and 100-120 min of sodium arsenite stress (supplementary figure S6c). Furthermore, we analyzed and quantified cytoplasmic patches observed for TDP-43^{Halo} and HaloTag alone (Figure 7 and supplementary figure S19). For TDP-43^{Halo} we were able to observe the formation of cytoplasmic TDP-43 patches, exhibiting confined TDP-43 with a strongly reduced mobility and an increasing size of these TDP-43 patches with sodium arsenite stress duration. A similar effect could not be observed for the HaloTag alone (supplementary figure S19), indicating the observed effect is not a general stress-related effect.

In the revised manuscript we present additional data where we further assessed the aspect of ATP depletion during sodium arsenite stress, to relate potentially reduced ATP levels with the observed decrease in TDP-43 mobility. We observed an insignificant reduction of ATP levels (supplementary figure S9), that are most likely not sufficient to explain the strong mobility reduction observed for TDP-43^{Halo} under sodium arsenite stress. It was also shown for other proteins that their mobility is not reduced

upon energy depletion e.g. (Munder et al., 2016), (Phair & Misteli, 2000), (Wagner, Chiosea, Ivshina, & Nickerson, 2004).

Point 3: The neuroglioma cell lines overexpress 33kDa HALO-tagged TDP-43 constructs that change the properties of the proteins. The C-terminal tag leads to a significant mis-localization of TDP-43 into the cytoplasm. This may be due to increased fragmentation and generation of truncated constructs missing the NLS and parts of the RRM domains. This physiological oligomerization of TDP-43 requires its N-terminal domain, and RNA binding requires the RRM1 & 2 domain. It is unclear from the data what specific or mixed protein species are being investigated in SGs and the cytoplasm. It is also unclear how the shift in mobility relates to pathology-relevant features such as RNA-binding, PTMs (phospho-TDP-43), fragmentation, and oligomerization/aggregation state of TDP-43.

Reply: We agree with the reviewer that for the C-terminally tagged cell line, we are not able to distinguish between full-length and fragmented TDP-43 in the course of the tracking analysis. However, we want to point out, that the N-terminally tagged TDP-43 cell line allows for the tracking of the full-length protein only. Supplementary figures S1 illustrates the establishment of the N-terminally tagged TDP-43 cell line. Tracking analysis using the N-terminally tagged cell line shows a similar stress-related and region specific reduction of TDP-43 mobility (Supplementary Figure S7). The TDP-43^{Halo} construct showed a slightly increased cytoplasmic mobility as compared to the HaloTDP-43 construct, indicating that a higher cytoplasmic mobility is most likely caused by fragmented TDP-43. We highlight in the revised version of the manuscript that the C-terminally tagged cell detects both truncated as well as full length TDP43.

Point 4: The Ctrl lanes in the Suppl. Figure S4 western blot for HALO-TDP-43 show a prominent 35kDa fragment, while the soluble and insoluble fractions do not.

Reply: The control lane is comprised of TDP-43^{Halo} cell lysates prepared by a different protocol and was used solely as a standard to compare different Western blots.

Finally, we would like to thank again the editor and all three reviewers for their thorough and constructive comments, which clearly helped us to improve our manuscript. We hope that it is now suitable for publication in the *Nature Communications*.

Sincerely yours,

Karin M. Danzer

Jens Michaelis

REVIEWERS' COMMENTS

Reviewer #1 (Remarks to the Author):

I would like to thank the authors for the significant efforts taken in revising the manuscript according to our comments, and I am fully satisfied by their revision.

I just have a minor comment regarding the analysis of the angular anisotropy of TDP-43 diffusion.

Here, to allow a fair comparison between diffusion in cytosol and in patches, the authors should first define ROIs of identical sizes in the two compartments, and then analyze the anisotropy for just the tracks that fall in these ROIs.

Reviewer #2 (Remarks to the Author):

The authors have adequately addressed my remaining concerns.

Reviewer #3 (Remarks to the Author):

The revised manuscript is certainly improved, by softening statements regarding oligomerizations. While this reviewer appreciates the effort Fig. S5, the result is still ambiguous with regard to any disease-relevant shift to aggregation.

Again, the authors have been responsive and made an effort to address questions about the stress response. While sorbitol treatment reduces mobility of TDP-43Halo, a similar change is observed for the tag alone. Certainly an improvement over the original submission, but still unclear how specific this is for TDP-43 or a very general effect on RNA-binding proteins in the cell.

As for the mislocalization/fragmentation of TDP-43, by far most data use the C-terminally tagged construct that shows partial cytoplasmic localization that is very different from the largely nuclear localization of the endogenous and N-terminally tagged protein. (A cytoplasmic TDP-43 splicing isoform has been reported, but it lacks the C-terminal prion-like domain).

In summary, the manuscript has been improved. While this study represents a major effort and has value in characterizing the behavior of tagged TDP-43 under stress conditions, there is remaining concern about the interpretation of data and the relevance for understanding TDP-43 proteinopathy-related changes.

As a more minor point, the term “bioreactors” in the abstract should be avoided, since it implies a site of high metabolic activity not relevant here. Perhaps this is a reference to "Stress granules as crucibles of ALS pathogenesis"? "Crucible" seems a much more suitable term, implying the formation of solid aggregates from liquid components.

Reply to Reviewers

We thank all reviewers for his/her careful evaluation of our work. We have considered the reviewer's comments and hereby submit a detailed point-by-point response for your consideration.

Reviewer #1 (Remarks to the Author):

I would like to thank the authors for the significant efforts taken in revising the manuscript according to our comments, and I am fully satisfied by their revision. I just have a minor comment regarding the analysis of the angular anisotropy of TDP-43 diffusion. Here, to allow a fair comparison between diffusion in cytosol and in patches, the authors should first define ROIs of identical sizes in the two compartments, and then analyze the anisotropy for just the tracks that fall in these ROIs.

Reply:

We thank the reviewer for his/her positive feedback. Following the reviewers' suggestion, we defined ROIs of identical sizes between cytosol, stress granules and patches analyzed the anisotropy of the jumps in the single-molecule tracking data. We added a statement regarding this additional analysis to the main text (page 13, lines 376-380) and show the analysis in new supplementary figure S22, with the methodology being explained in the Figure caption.

Reviewer #2 (Remarks to the Author):

The authors have adequately addressed my remaining concerns.

Reply:

We thank the reviewer for his/her work and effort.

Reviewer #3 (Remarks to the Author):

The revised manuscript is certainly improved, by softening statements regarding oligomerizations. While this reviewer appreciates the effort Fig. S5, the result is still ambiguous with regard to any disease-relevant shift to aggregation. Again, the authors have been responsive and made an effort to address questions about the stress response. While sorbitol

treatment reduces mobility of TDP-43Halo, a similar change is observed for the tag alone. Certainly an improvement over the original submission, but still unclear how specific this is for TDP-43 or a very general effect on RNA-binding proteins in the cell. As for the mislocalization/fragmentation of TDP-43, by far most data use the C-terminally tagged construct that shows partial cytoplasmic localization that is very different from the largely nuclear localization of the endogenous and N-terminally tagged protein. (A cytoplasmic TDP-43 splicing isoform has been reported, but it lacks the C-terminal prion-like domain). In summary, the manuscript has been improved. While this study represents a major effort and has value in characterizing the behavior of tagged TDP-43 under stress conditions, there is remaining concern about the interpretation of data and the relevance for understanding TDP-43 proteinopathy-related changes. As a more minor point, the term “bioreactors” in the abstract should be avoided, since it implies a site of high metabolic activity not relevant here. Perhaps this is a reference to “Stress granules as crucibles of ALS pathogenesis”? “Crucible” seems a much more suitable term, implying the formation of solid aggregates from liquid components.

Reply:

We thank the reviewer for his/her positive words that “*the manuscript has certainly improved*” and his/her appreciation of the new Fig. S5. We also acknowledge the constructive criticism of the reviewer. Following the reviewer’s suggestion we changed the term “bioreactor” the abstract and refer to “stress granules crucibles of ALS pathogenesis”. We fully agree with the reviewer that this term brings this better to the point.